# Post-Operative Anorectal Manometry in Children following Anorectal Malformation Repair: A Systematic Review

**DOI:** 10.3390/jcm12072543

**Published:** 2023-03-28

**Authors:** Hannah M. E. Evans-Barns, Melissa Y. Tien, Misel Trajanovska, Mark Safe, John M. Hutson, Phil G. Dinning, Sebastian K. King

**Affiliations:** 1Department of Paediatric Surgery, The Royal Children’s Hospital, 50 Flemington Road, Parkville, VIC 3052, Australia; hannah.evans-barns@rch.org.au (H.M.E.E.-B.); misel.trajanovska@mcri.edu.au (M.T.); 2Surgical Research Group, Murdoch Children’s Research Institute, 50 Flemington Road, Parkville, VIC 3052, Australia; john.hutson@mcri.edu.au; 3Department of Paediatrics, University of Melbourne, Melbourne, VIC 3052, Australia; 4Department of Gastroenterology and Clinical Nutrition, The Royal Children’s Hospital, 50 Flemington Road, Parkville, VIC 3052, Australia; mark.safe@rch.org.au; 5Department of Urology, The Royal Children’s Hospital, 50 Flemington Road, Parkville, VIC 3052, Australia; 6Department of Surgery, College of Medicine and Public Health, Flinders University and Flinders Medical Centre, Bedford Park, SA 5042, Australia; phil.dinning@flinders.edu.au

**Keywords:** manometry, anorectal malformation, gastrointestinal motility, high-resolution manometry, anorectum

## Abstract

Despite surgical correction, children with anorectal malformations may experience long-term bowel dysfunction, including fecal incontinence and/or disorders of evacuation. Anorectal manometry is the most widely used test of anorectal function. Although considerable attention has been devoted to its application in the anorectal malformation cohort, there have been few attempts to consolidate the findings obtained. This systematic review aimed to (1) synthesize and evaluate the existing data regarding anorectal manometry results in children following anorectal malformation repair, and (2) evaluate the manometry protocols utilized, including equipment, assessment approach, and interpretation. We reviewed four databases (Embase, MEDLINE, the Cochrane Library, and PubMed) for relevant articles published between 1 January 1985 and 10 March 2022. Studies reporting post-operative anorectal manometry in children (<18 years) following anorectal malformation repair were evaluated for eligibility. Sixty-three studies were eligible for inclusion. Of the combined total cohort of 2155 patients, anorectal manometry results were reported for 1755 children following repair of anorectal malformations. Reduced resting pressure was consistently identified in children with anorectal malformations, particularly in those with more complex malformation types and/or fecal incontinence. Significant variability was identified in relation to manometry equipment, protocols, and interpretation. Few studies provided adequate cohort medical characteristics to facilitate interpretation of anorectal manometry findings within the context of the broader continence mechanism. This review highlights a widespread lack of standardization in the anorectal manometry procedure used to assess anorectal function in children following anorectal malformation repair. Consequently, interpretation and comparison of findings, both within and between institutions, is exceedingly challenging, if not impossible. Standardized manometry protocols, accompanied by a consistent approach to analysis, including definitions of normality and abnormality, are essential to enhance the comparability and clinical relevance of results.

## 1. Introduction

Anorectal malformations represent a spectrum of anomalies affecting the anus, rectum, urinary, and/or genital tracts [1]. The fundamental aims of surgical correction remain consistent today with those described by Rudolph Matas in 1897: “the ideal result of this kind of operation is the restoration of the passage of stool, creating an anus in a normal position with bowel control” [2]. Outcomes with respect to bowel function have greatly improved alongside the evolution of operative repair techniques, most notably, following the advent of the posterior sagittal anorectoplasty [PSARP]) [3]. However, persistent bowel dysfunction, including constipation and fecal incontinence, continues to impact upon a significant proportion of patients long-term [4,5].

As the anal sphincter plays a critical role in both fecal continence and defecation, its function in children with persistent bowel problems after surgical procedures becomes a focal point for investigation. Anorectal manometry is the most widely used investigation to identify abnormalities of anorectal coordination and/or anal sphincter complex dysfunction [6]. The assessment typically comprises a combination of pressure measurements, including evaluation of involuntary function of the anal canal (at rest); voluntary function (squeeze); rectal balloon distension to determine the existence of the rectoanal inhibitory reflex (RAIR); rectoanal coordination during simulated defecation (push maneuver); and rectal sensation [6,7,8,9]. 

Despite extensive testing with anorectal manometry, the relationships between manometry results and patient symptoms remain poorly defined. In this review, we sought to summarize the methodology and outcomes of anorectal manometry performed in children following repair of anorectal malformations, to appraise current understanding of anorectal function, and guide future work in this cohort.

## 2. Methods

This systematic review was conducted in compliance with Preferred Reporting Items for Systematic Reviews and Meta-Analyses (PRISMA) [10]. A primary search was conducted in Embase, MEDLINE, PubMed, and the Cochrane Library in March 2020, and subsequently repeated in March 2022. The search was restricted to human studies published since 1st January 1985. The methodology was published prospectively on PROSPERO (PROSPERO registration: CRD42020177344). The search strategy is summarized in Appendix B.

### 2.1. Study Selection

After removal of duplicate articles, title and abstract were assessed for eligibility independently by two authors (H.E.B. and M.Y.T.). Cohort studies, case studies, longitudinal studies, and clinical trials were included for review. Conference abstracts, meta-analyses, systematic reviews, animal studies, and in vitro studies were excluded.

The following inclusion criteria were utilized for abstract screening: anorectal manometry performed in children following surgical repair of an anorectal malformation, published in the English language. Studies reporting manometry outcomes in mixed populations (anorectal malformations and other conditions, mixed pediatric and adult cohorts) were included, provided results from children (aged 18 years or less) with anorectal malformations were reported separately in the final analysis. Studies reporting only the pre-operative use of anorectal manometry were excluded.

### 2.2. Data Extraction

Data were independently extracted by two authors (H.E.B. and M.Y.T.). Extracted cohort characteristics included patient sex; age at assessment; anorectal malformation type; associated anomalies; and post-operative bowel function, including assessment instruments and outcomes. With respect to manometry characteristics, extracted data points included manometry type, including catheter specifications; assessment protocol, including parameters assessed; and motility outcomes, including bowel function correlates.

### 2.3. Quality Assessment

Non-randomized studies were appraised using the Newcastle–Ottawa Scale [11]. The scale consists of eight items, which evaluate methodological quality based on criteria related to selection, comparability, and exposure. A maximum total of nine may be awarded to the highest quality studies [11].

## 3. Results

### 3.1. Search Results

A total of 283 unique articles were identified after removal of duplicate records. Following full-text review, 63 articles were identified, which reported findings of anorectal manometry in children with repaired anorectal malformations. Search results and study selection are presented in Figure 1.

From the combined total cohort of 2155 patients, anorectal manometry results were reported for 1755 children (age range 2 months–18 years) with repaired anorectal malformations. The remainder were either part of a study with a mixed diagnostic cohort (and had a condition other than an anorectal malformation), and/or only a proportion of the cohort underwent anorectal manometry. The median manometry cohort size was 22 children (range 5–115). Insufficient data were provided to calculate the median age or sex ratio. Study characteristics are summarized in Table 1.

Anorectal malformation type was specified for the majority of the cohort (1523/1755 children, 86.8%). The Krickenbeck and Wingspread anorectal malformation classification systems were the most frequently utilized [12,13]. The most frequently reported malformation types were, “high” (267 children); “rectoprostatic fistula” (198 children); and “intermediate” (169 children). The approach to operative repair type was reported for 1250/1755 (71.2%) children in the manometry cohort. Of these, the posterior sagittal anorectoplasty (PSARP) and its variants were the most commonly utilized procedures (868/1250, 69.4%), followed by laparoscopically assisted anorectoplasty (LAARP) (73/1250, 5.8%). Details regarding associated malformations (including sacrospinal anomalies) were provided by less than half of the studies identified for review (28/63, 44.4%). Cohort clinical characteristics are summarized in Table 1.

Studies utilized anorectal manometry to evaluate a range of aspects relating to the management of anorectal malformation patients. Anorectal manometry was performed to compare malformation types [14,15,16,17,18,19]; appraise operative techniques [20,21,22,23,24,25,26,27,28,29,30,31,32,33,34,35,36,37,38,39]; evaluate post-operative assessment modalities [40,41,42,43,44,45,46,47,48,49,50,51,52,53,54,55]; assess and/or prognosticate bowel function [45,56,57,58,59,60]; and investigate the pathophysiology [46,61,62,63,64] and management [65,66,67,68,69,70,71,72] of post-operative bowel dysfunction.

### 3.2. Quality Assessment

Only 12 included studies adequately addressed the criteria outlined in the Newcastle Ottawa Scale and were classified as “good quality”. The majority were classified as “poor quality”, predominately due to limitations identified in the “comparability” category. Quality evaluations are presented in Appendix A.

**Table 1 jcm-12-02543-t001:** Clinical characteristics of included studies.

First Author	Year	Cohort (Total (Male))		Study Population Summary	Age at Time of Manometry	Reported Anorectal Malformation Type ^1^	Associated Anomalies	Surgical Repair Type
		**Total ^2^**	**Manometry ^3^**					
Arnoldi [14]	2014	30 (11)	30 (11)	Toilet-trained children with anorectal malformation with a good, predicted prognosis ^4^	Range: 2.5–10 years. Measure of central tendency not provided. Mean follow-up at assessment: 5 years.	Rectoperineal fistula: 19 Rectovestibular fistula: 10 Imperforate anus: 1	Tethered cord: 6/30 (20%) Excluded: abnormal sacrum. Other anomalies not reported.	Three-stage repair (diverting colostomy, PSARP, colostomy closure): 9 (30%) Primary PSARP: 21/30 (70%)
Banasiuk [64]	2021	12 (-)	12 (-)	Children who had undergone surgery for anorectal disorders, including anal atresia	Median 70 (16–195) months	Perineal fistula (7) Rectourethral fistula (4) Persistent cloaca (1)	Not reported	Not reported
Becmeur [23]	2001	14 (9)	10 (-)	Children following three-flap anoplasty for primary or re-do repair	Not reported. Mean age at study conclusion: 6 years (range 3–14).	High/intermediate: rectobulbar fistula (8); rectovesical fistula (1); cloacal defect (1); long rectal atresia (1). Low: rectovulvar fistula (3). Not specified for manometry cohort	T21 (1); HD (1); psychomotor troubles (2); complex caudal malformation (3); GI duplication (1); ductus arteriosus (1); crossed renal ectopia (1); renal cystic dysplasia (1); hypospadias (1); supernumerary hemivertebra (1).	Three flap anoplasty: 14/14 Primary repair: 9/14 Re-do procedure: 5/14
Bhat [56]	2008	10 (9)	10 (9)	High or intermediate anorectal malformation, following sigmoid colostomy formation but prior to PSARP	Post-PSARP age: mean 26.3 months (range 15–57).	Rectoprostatic fistula: 6 Rectobulbar fistula: 3 Rectovaginal fistula: 1	Not reported	PSARP: 10/10
Burjonrappa [61]	2010	86 (53)	6/14 (4) ^5^	Patients with megarectum following surgery for anorectal malformation	Not reported.	High: 23 -developed megarectum: 6 (26%) Low: 63 -developed megarectum: 8 (13%)	Sacral vertebral anomalies: 13/86	Children with megarectum: Mollard anterior approach: 1 Cutback: 1 Anal transposition: 2 Laparoscopic primary pullthrough: 3 Posterior-sagittal pullthrough: 2 Y-V plasty: 2 Dilatation only: 2
Cahill [24]	1985	6 (6)	5 (5)	Patients with anorectal malformation following PSARP	Mean: 2.2 years (range 1.25–3.5 years)	Rectoprostatic fistula: 6	Sacral anomaly: 1	PSARP: 6/6
Caldaro [43]	2012	17 (13)	17 (13)	Neurologically healthy children, >4 years, with constipation/FI, following anorectal malformation repair	Mean: 8.3 years (range 5–15 years).	High: rectobladderneck fistula (2); rectourethral fistula (4); cloaca (1). Intermediate: rectourethral fistula (4). Low: rectoperineal fistula (2); rectovestibular fistula (3); anal stenosis (1).	Myelomeningocele: 2 Tethered cord: 1 Renal: 2 VACTERL syndrome: 1	PSARP: 17/17
Caruso [67]	2015	25 (15)	25 (15)	Neurologically heathy children >4 years with “true” FI following anorectal malformation repair	Mean: 6.5 years (range 5–9 years).	Vestibular fistula: 3 Rectal atresia: 8 Bulbar fistula: 5 Prostatic fistula: 2 Cloaca: 2 Vaginal fistula: 1 Vesical fistula: 4	Renal: 12 Genital: 10 Spinal: 5	PSARP: 25 Laparoscopically assisted proportion not defined.
Caruso [72]	2021	14 (-)	14 (-)	Children with FI or bowel dysfunction not responsive to conventional laxative treatment, receiving transanal irrigation	Mean 10.29 ± 3.25 years	Rectobulbar fistula: 4 Rectovaginal fistula: 2 Rectovesical fistula: 4 Rectoprostatic fistula: 2 Perineal fistula: 2	Spinal: 6	Not reported
Chen [73]	1998	58 (34)	44 (-)	All children with anorectal malformation repaired by a single surgeon	Not reported. Manometry was performed within the 1^st^ year of the repair and repeated at 1–2-year intervals.	High/intermediate. Blind pouch (10); fistula: rectobulbar (4); rectoprostatic (7); rectovesical (2); rectovaginal (1); rectovestibular (4); cloacal anomaly (2). Low. Fistula: anocutaneous and anterior perineal anus (16); rectoperineal (4); anovulvar or anovestibular (8). Not specified for manometry cohort	High/intermediate: EA (1); T21 (2); spinal (2); urogenital (5); cardiac (3). Low: EA (1); urogenital anomaly (2); duodenal obstruction (1); cardiac (2).	LSARP: 32 PSARP: 29 R-ASPA: 5 Total surgical cohort (*n* = 66). Operative details of study (*n* = 58) and manometry (*n* = 44) cohorts not provided.
Chung [22]	2018	30 (-)	30 (-)	Toilet-trained children following PSARP or LAARP without neurological comorbidities or cloacal malformation	PSARP: 15.5 years (range 8–32) LAARP: 9 years (range 5–14)	Krickenbeck classification: (PSARP, LAARP) Rectovesical: 2, 3 Rectoprostatic: 7, 8 Rectobulbar: 5, 5	Not reported	PSARP: 14/30 (46.7%) LAARP: 16/30 (53.3%)
Doolin [44]	1993	25 (15)	25 (15)	Children following repair of anal atresia	Mean: 9.6 years (range 6–16)	Rectourethral fistula: 15 Rectovaginal fistula: 5 Vestibular fistula: 1 Cloaca: 3 Anorectal malformation without fistula: 1	Not reported	Abdominoperineal pullthrough: 20 Presacral pullthrough: 4 Perineal anoplasty: 1
El-Debeiky [25]	2009	15 (15)	9 (9)	Males with high anorectal malformation treated with laparoscopic-assisted pull-through	Not reported. Assessed at 3 years of age or older.	Rectobladderneck fistula: 15/15	Not reported	Laparoscopically assisted pull-through: 15/15
Emblem [40]	1994	16 (8)	16 (8)	Adolescents with low anorectal malformations following repair	Mean: 14.9 years (12–16)	Anocutaneous fistula: 16 (100%)	Not reported	Anal dilatation: 5 Anal cutback: 11 Secondary procedures: Anal transposition: 3 Posterior anoplasty: 2
Emblem [45]	1997	33 (16)	33 (16)	Adolescents with anorectal malformations following repair	Intermediate/high: mean 15.8 years (14.6–17.1) Low: mean 14.9 years (13.7–16.2)	High/intermediate: 16 (48%) Low: 17 (52%)	Not reported	Sacroperineal, sacroabdominoperineal, abdominoperineal, perineal procedures according to the malformation type. Cohort distribution not reported.
Fukata [41]	1997	15 (14)	15 (14)	Patients with high or intermediate anorectal malformations	Median: 14 years (8–18)	High: 10 (67%) Intermediate: 5 (33%)	Not reported	Abdominoperineal rectoplasty: 15 (100%)
Hedlund [46]	1992	30 (-)	30 (-)	Patients with anorectal malformations following PSARP, without major sacral malformation	Range: 5–18 years. Measure of central tendency not provided	Bulbar fistula: 9 Prostatic fistula: 8 Vaginal fistula: 1 Vestibular fistula: 6 Rectal atresia: 3 No fistula: 3	Not reported	PSARP: 30
Heikenen [74]	1999	13 (6)	Colonic manometry: 13 (6) Anorectal manometry: 10 (4)	Children with FI refractory to standard medical therapy following repair of anorectal malformation	Mean 8.6 years (range 5–13)	High: 7/10 Low: 3/10	Not reported	Not reported
Hettiarachchi [47]	2002	15 (5)	15 (5)	Children with chronic constipation and/or FI following anorectal malformation repair	Median 2.5 years (range 1–15)	High: 6 Intermediate: 5 Low: 4	Minor sacroneural anomalies: 3 Partial trisomy 22: 2	“Variety of techniques used for reconstruction” [47]; not further specified.
Huang [20]	2017	89 (0)	43 (0)	Female patients with rectovestibular fistula	Not reported. Age at operation: -Modified semi-PSARP: 1.6 months (9 days–2.5 months) -Transperineal anal transposition: 6.4 (5–8) months Anorectal manometry performed 12 months post-operatively.	Imperforate anus with rectovestibular fistula: 43	Cardiac: 27 Renal: 10 OA: 2 Polydactyly: 1 Absent coccyx: 1 Sacrococcygeal pilonidal sinus: 1	One-stage modified semi-PSARP: 39 Transperineal anal transposition: 50 Manometry recipients: 43/89 (48% total cohort) -Modified semi-PSARP: 17 (39%) -Transperineal anal transposition: 26 (61%)
Husberg [26]	1992	48 (24)	43 (-)	Children with high or intermediate anorectal malformation	Range 7 months–16 years	Rectovesical fistula: 1 Rectoprostatic fistula: 15 Rectobulbar fistula: 4 Blind-ending rectum: 5 Cloacal malformation: 11 Rectovaginal fistula: 1 Rectovestibular fistula: 11 Not specified for manometry cohort	Concomitant malformations: 44	Posterior sagittal approach: 48
Ishihara [48]	1987	49 (-)	49 (-)	Patients with anorectal malformations following repair	Mean 9.5 years (range 5–20 years ^6^).	Translevator: 9 Intermediate: 6 Supralevator: 24 Operated at another hospital: 10	Not reported	Not reported
Iwai [15]	1988	28 (-)	17 (-)	Patients with anorectal malformations following repair	Range: 5–14 years. Measure of central tendency not provided	High: 13 Intermediate: 6 Low: 9 Not specified for manometry cohort	Not reported	Not reported. Department practice -high/intermediate: abdominoperineal rectoplasty -low: perineoplasty
Iwai [68]	1993	8 (-)	8 (-)	Children undergoing biofeedback training for FI following anorectal malformation repair	Range: 6–12 years. Measure of central tendency not provided	High: 6 Intermediate: 2	Not reported	Abdominoperineal rectoplasty: 7 Abdominosacroperineal rectoplasty: 1
Iwai [65]	1997	14 (-)	14 (-)	Children undergoing biofeedback training for FI following anorectal malformation repair	Range: 5–14 years. Measure of central tendency not provided	High: 12 Intermediate: 2	Sacral deformity: 0	Not reported
Iwai [66]	2007	5 (-)	5 (-)	Children with severe constipation following anorectal malformation repair, treated with herbal medication	Mean: 11.5 years (range 7–17 years)	High: 4 Cloacal anomaly: 1	Not reported	Not reported
Keshtgar [69]	2007	16 (7)	16 (7)	Children undergoing excision of megarectum for intractable FI	Median: 9 years (range 2–15 years)	High: 6 Intermediate: 4 Low: 6	EA: 3 Duodenal atresia: 1 Caudal regression with sacral dysgenesis: 1 T21: 1	Various approaches described for wider population; detail not provided for manometry cohort.
Keshtgar [49]	2008	54 (27)	54 (27)	Children with intractable FI/constipation following anorectal malformation repair	High (*n* = 34) ^7^ Median: 10.5 years (range 3.9–21.8) Low (*n* = 20) Median: 9.4 years (range 4.1–15.3).	Low: anal stenosis and anterior ectopic anus (12); perineal fistula (8). High: rectovestibular fistula (12); rectovesical (2); cloacal malformation (1); rectoprostatic urethral fistula (19).	Megarectum (16); megarectum + neuropathy (11); neuropathy (8); caudal regression (2); sacral dysgenesis (1); esophageal atresia (9); HD (1); T21 (1); T22 (1).	PSARP: 24 Abdominoperineal pullthrough: 8 Stephens: 2 Mollard: 1 Durham Smith: 3 Anal transposition: 7 Anoplasty: 9
Kimura [27]	2010	28 (25)	28 (25)	Infants managed for high anorectal malformation	Not reported	Rectoprostatic fistula: 21 No fistula: 3 Rectovaginal fistula: 2 Rectovesical fistula: 1 Cloacal malformation: 1	Not reported	Open ARP: 15 Laparoscopic ARP: 13
Kudou [28]	2005	20 (-)	20 (-)	Children following LAARP or PSARP for management of high anorectal malformation	LAARP: mean 50.5 months (SD 10 months) PSARP: mean 73 months (SD 12 months).	LAARP: rectourethral fistula (7); rectovesical fistula (2); no fistula (1); rectovaginal fistula (2); cloaca (1). PSARP: rectourethral fistula (2); rectovesical fistula (2); no fistula (1); cloaca (2).	Spinal lipoma: 2	LAARP: 13 PSARP: 7
Kumar [57]	2010	32 (18)	32 (18)	Children with anorectal malformations following repair	Infant group: mean 8 months (range 6–12 months) Child group: mean 3.4 years (range 15 months–5.9 years)	Rectoprostatic fistula: 5 Rectobulbar fistula: 3 No fistula: 3 Rectovestibular fistula: 2 Rectovaginal fistula: 1 Anocutaneous fistula: 7 Rectoperineal fistula: 2 Anovestibular fistula: 9	Renal: 13 Cardiac: 16 Sacral: 4 VACTERL: 1 Chromosomal: 1	PSARP (high anomalies) or anoplasty (low anomalies).
Langemeijer [29]	1991	50 (28)	39 (-)	Patients with high malformation following PSARP	Not reported. Age at operation: -primary PSARP: 1 month–5 years -redo PSARP: 6–16 years	High: 39	Not reported	Primary PSARP: 40 Redo PSARP: 10 Not specified for manometry cohort
Leung [70]	2006	12 (10)	9 (-)	Children >5 years old with FI following anorectal malformation repair	Age range: 5–17 years during program completion.	High: 7 Intermediate: 5 Not specified for manometry cohort	Not reported	Pullthrough: 7 PSARP: 5
Lin [30]	1996	27 (-)	27 (-)	Children with high or intermediate malformation following PSARP or R-APSA	PSARP: <4 years: 10 >4 years: 13 R-APSA: <4 years: 4	Rectoprostatic fistula: 8 Rectobulbar fistula: 3 Rectovaginal fistula: 3 Blind: 11 Cloacal anomaly: 2	Spinal: 1 T21: 1 Renal: 2 Genital: 3 Esophageal: 1	PSARP: 23 R-APSA: 4
Lin [31]	2003	22 (16)	22 (16)	Patients with high or intermediate malformations following LAR or PSARP	LAR group: mean 16.2 months (range: -) PSARP: mean 17.1 months (range: -)	Rectoprostatic fistula: 12 Rectobulbar fistula: 3 Rectovaginal fistula: 6 Blind pouch: 1	T21: 2 Genital: 3 Cardiac: 2 Renal: 1 Esophageal: 1	LAR: 9 PSARP: 13
Liu [32]	2004	113 (90)	113 (90)	Patients with intermediate or high malformations following PSARP	Not reported. Median age at last follow up: -Group 1: 7.8 years (range 4.9–13 years) -Group 2: 6.7 years (range 5.3–11.6 years)	Rectovesical fistula: 5 Rectoprostatic fistula: 10 Anorectal agenesis w/o fistula: 15 Rectovaginal fistula: 1 Rectobulbar fistula: 49 Anal agenesis w/o fistula: 12 Rectovestibular fistula: 21	Sacral anomalies: 6 T21: 2 Genital: 4	Traditional PSARP: 48 One-stage PSARP: 65
Martins [50]	1996	27 (17)	27 (17)	Patients with intermediate or high malformations following PSARP	Range: 4–11 years. Measure of central tendency not provided.	Rectourethral fistula: 17 Rectovaginal fistula: 5 Rectovestibular fistula: 5	Sacral malformation: 10	PSARP: 27
Mert [55]	2021	23 (18)	23 (18)	Children following anorectal malformation repair, able to cooperate during anorectal manometry without neurological or neurosurgical disorders	Median: 7 (range 5–14) years	Anorectal agenesis without fistula: 5 Rectovesical fistula: 5 Rectobulbar fistula: 4 Rectovestibular fistula: 2 ^8^ Rectoperineal fistula: 7	Not reported	Not reported
Mollard [75]	1991	21 (-)	13 (-)	Patients with intermediate or high malformations following repair	Not reported	High: 7 Intermediate: 6	Not reported	Anterior perineal approach
Nagashima [62]	1992	159 (108)	32 (-)	Children following repair of anorectal malformations	Mean: 9 years (range 5–16 years).	High: 17 Intermediate: 6 Low: 9	Not reported	Abdominoperineal rectoplasty: high/intermediate (23) Perineoplasty: low (9)
Niedzielski [33]	2008	94 (44)	91 (42)	Children following PSARP	Not reported. Assessed 6 months–16 years following PSARP (mean 11.4 years).	Not specified for manometry cohort Perineal fistula (26); vestibular fistula (17); vaginal fistula (5: 2 low, 3 high); anal stenosis (3); cutaneous fistula (1); cloaca (4); no fistula (10); midline raphe fistula (6); bulbar fistula (5); prostatic fistula (9); rectoanal stenosis (5); bladder neck fistula (2); rectal atresia (1). Specified for manometry cohort: high (68); low (23).	Not reported	PSARP: 91
Ninan [71]	1994	13 (9)	13 (9)	Children undergoing levatorplasty for management of FI following anorectal malformation repair	Not reported. Age at levatorplasty: mean 10.7 years (4–17)	High malformation: 11 Rectovestibular fistula: 1 Cloaca: 1	Renal: 6 Cardiac: 1 Esophageal: 2 T21: 1 Spinal: 1	Sacroperineal pullthrough: 5 Sacroabdominoperineal pullthrough: 7 Anoplasty: 1
Okada [21]	1993	10 (4)	6 (-)	Patients following ASARP for re-operation due to FI following anorectal malformation repair	Not reported. Median age at redo operation: 4.5 years (range 2–7). Median follow-up at assessment: 3.29 years (range 1.1–5.3).	High: 3 Intermediate: 5 Low: 2 Not specified for manometry cohort	Not reported	ASARP (re-operation): 10/10
Penninckx [16]	1986	54 (20)	19 (-)	Infants with anorectal malformation treated at a single center	Vaginal anal canal: mean 31 ± 15 months Urethral anal canal: mean 7 ± 3 months Vesical anal canal: mean 43 ± 6 months No orifice: mean 14 ± 15 months	Vaginal anal canal: 6 Urethral anal canal: 5 Vesical anal canal: 2 No orifice: 6	Not reported	Variety of repair strategies reported, without specification of cohort size.
Ray [76]	2004	115 (69)	115 (69)	Children with intermediate or high malformation, following PSARP	Not reported.	High: 12 Intermediate: 103	Spinal: 2 Renal: 3	PSARP: - ASARP: -
Ren [34]	2019	48 (48)	22 (22)	Children with intermediate malformations following SILAARP or PSARP	SILAARP: 29.20 months ± 10.21 PSARP: 32.07 months ± 10.54	Rectobulbar fistula: 48	Spinal: 18 Renal: 16 Cardiac: 11	PSARP: 14 SILAARP: 34
Rintala [35]	1990	30 (25)	30 (25)	Patients with intermediate or high malformations following repair	Group 1: mean 3.1 years (range 1–6 years) Group 2: mean 8.8 years (range 5–13 years)	High: 25 Intermediate: 5	Not reported	Group 1: 14 PSARP (12) or sacroperineal pull-through (2). Group 2: 16 primary sacro-abdominoperineal pull-through (16); with nine undergoing secondary reconstruction by PSARP (9).
Rintala [36]	1995	65 (36)	53 (-)	Patients with intermediate or high malformations	Not reported	Rectoprostatic fistula: 26 Rectobulbar fistula: 5 Anal agenesis: 4 Rectovesical fistula: 2 Cloaca: 9 Rectovestibular fistula: 16 Rectovaginal fistula: 3 Not specified for manometry cohort	Not reported	PSARP: 53
Rintala [58]	1993	40 (22)	40 (22)	Patients with intermediate or high anorectal malformations	Not reported	Rectoprostatic fistula: 17 Rectobulbar fistula: 4 Anal agenesis: 2 Cloaca: 8 Rectovestibular fistula: 8 Rectovaginal fistula: 1	Not reported	PSARP: 40
Rintala [17]	1990	10 (-)	9 (-)	Patients with intermediate or high malformations	Not reported	High: - Intermediate: -	Not reported	PSARP: 9
Rintala [59]	1995	46 (25)	46 (25)	Patients following PSARP for intermediate and high malformations	Mean 6.2 years (range 3.8–10).	Rectoprostatic fistula: 19 Rectobulbar fistula: 4 Anal agenesis: 3 Cloaca: 9 Rectovestibular fistula: 9 Rectovaginal fistula: 2	Significant sacral/spinal defects: 11 T21: 2	PSARP: 46
Rintala [60]	1995	16 (14)	16 (14)	Patients undergoing secondary PSARP for intractable FI following primary anorectal malformation repair	Not reported. Manometry performed prior to secondary repair. Age at secondary repair: mean 12.4 years (range 8–16).	Rectourethral fistula: 11 Rectovesical fistula: 2 High anomaly, no fistula: 1 Cloaca: 2	Significant sacral anomalies: 4	Secondary repair PSARP: 16 Primary repair Sacroabdominoperineal pullthrough: 10 Abdominoperineal pullthrough: 6
Ruttenstock [18]	2013	12 (0)	12 (0)	Patients with an externally accessible fistula	Median 585 days (range 197–1287 days)	Perineal fistula: 7 Rectovestibular fistula: 5	Cardiac: 3 Renal: 4 Sacral: 2 Chromosomal: 1	ASARP: 6 Mini-PSARP: 6
Sangkhathat [63]	2004	24 (18)	24 (18)	Infants less than three years of age, post-anoplasty for treatment of anorectal malformation	Mean 9.6 months (range 2–36 months)	Perineal fistula: 6 Vestibular fistula: 3 Rectobulbar urethral fistula: 6 Rectovaginal fistula: 1 Blind rectal pouch: 8	T21: 4 Opitz syndrome: 1 Caudal regression: 1	PSARP: - Posterior myectomy and Y-V plasty: - Limited PSARP: -
Schuster [19]	2000	10 (1)	10 (1)	Patients managed for perineal fistula using anal transposition technique	Mean 20.75 months (range 6–72 months)	Perineal fistula: 10	Not reported	Anal transposition technique: 10
Schuster [42]	2001	17 (10)	17 (10)	Children with anorectal malformations following PSARP	Mean: 5.4 years (32–120 months)	Fistula location: rectovaginal (3); vestibular (2); rectovesical (2); prostatic (1); bulbar urethral (4) perineal (2). Cloacal malformation: 2 Rectal atresia: 1	Normal sacrum: 7/17 (42%)	PSARP: 17/17 (100%) Secondary PSARP: 4/17 (24%)
Senel [51]	2007	18 (12)	18 (12)	Children with anorectal malformations following repair	Mean 6.2 years (range 63–104 months)	Rectovesical fistula: 1 Rectoprostatic fistula: 4 Rectal atresia: 1 Rectovestibular fistula: 5 Rectovaginal: 1 ^9^ Rectobulbar: 6	Not reported	PSARP: 9 Sacroperineal pullthrough: 3 Perineal pullthrough: 6
Sonnino [37]	1991	5 (3)	5 (3)	Children with FI treated with gracilis muscle transposition	Mean: 13 years (range 10–17 years)	High anorectal malformation: 5	VACTERL: 3	Gracilis muscle transposition: 5 Primary repair not reported.
Tang [52]	2017	6 (2)	6 (2)	Children with defecation disorders	Mean 5 years (range 4–9)	Not reported	Not reported	PSARP: 6
Tong [38]	2011	61 (50)	61 (50)	Infants with high anorectal malformation treated with LAARP vs. PSARP	3.1–4.4 years Age at operation: LAARP: mean 5.3 months (range 3–10) PSARP: mean 4.9 months (range 3–11)	Rectoprostatic fistula: 39 Rectobulbar fistula: 13 Rectovesical fistula: 2 Rectovaginal fistula: 7	Not reported	LAARPT: 33 PSARP: 28
Vital Junior [53]	2007	82 (46)	82 (46)	Children with anorectal malformation following PSARP	Mean: 85.5 months (range 12–204)	High: 45 Intermediate: 37	Not reported	PSARP: 82
Wang [54]	2016	47 (31)	47 (31)	Children treated at a single center without congenital megarectum, sacral or spinal deformities	Mean: 4 years (range 1.4–8.9)	Rectourethral fistula: 15 Rectovaginal fistula: 2 Rectovesical fistula: 1 Perineal fistula: 11 Anovestibular fistula: 6 Anal stenosis: 2 Anal hole ^10^: 1 No fistula: 9	Not reported	PSARP: 23 Transperineal anorectoplasty: 24
Yang [39]	2009	23 (19)	23 (19)	Children with high anorectal malformations following PSARP or LAARP	LAARP: mean 17.4 months (SD 4.9) PSARP: mean 19.3 months (SD 6.2)	Rectourethral fistula: 6 Rectoprostatic fistula: 5 Rectovesical fistula: 4 Rectovaginal fistula: 3 Anorectal agenesis: 5	Not reported	LAARPT: 11 (11 male) PSARP: 12 (8 male)

^1^ Anorectal malformation type, as reported in original article. ^2^ Total cohort with anorectal malformation. ^3^ Refers to cohort of children undergoing post-operative anorectal manometry following anorectal malformation repair. ^4^ Good prognosis: specific malformation types (rectoperineal fistula, rectovestibular fistula, imperforate anus without fistula, rectal atresia, cloaca with common channel < 3 cm); associated with a prominent midline groove, suggestive of good perineal muscle, and a normal sacrum. ^5^ Six of 14 patients with megarectum underwent post-operative anorectal manometry. ^6^ Subgroup of children operated on at another hospital with normal bowel function excluded due to age >18 years. Other analyses included in the narrative synthesis. ^7^ Findings of the “high” malformation group excluded from synthesis due to inclusion of participants >18 years of age. ^8^
Table 1 listed a total of 29 patients, including eight patients following repair of rectovestibular fistula, which is greater than the total study number reported (*n* = 23). Table 2 (*n* = 23) reported two patients with rectovestibular fistulae: data taken from Table 2. ^9^ The number of patients reported to have undergone repair of rectovaginal fistulae differed in the article; data included as presented in Table 1 and Table 2. ^10^ Not further specified. ARP: abdominoperineal rectoplasty; ASARP: anterior sagittal anorectoplasty; CM: colonic manometry; EA: esophageal atresia; FI: fecal incontinence; GI: gastrointestinal; HD: Hirschsprung disease; LAARP: laparoscopically assisted anorectoplasty; LAARPT: laparoscopically assisted anorectal pullthrough; LAR: laparoscopically assisted pull-through anorectoplasty; LSARP: limited sagittal anorectoplasty; PSARP: posterior sagittal anorectoplasty; R-APSA: Rehbein’s mucosa-stripping endorectal pull-through; R-ASPA: Rehbein’s mucosa-stripping endorectal pull-through in combination with anterior sagittal perineal anorectoplasty; SILAARP: single-incision laparoscopic-assisted anorectoplasty; SD: standard deviation; T21: trisomy 21; T22: trisomy 22; VACTERL: Vertebral, Anorectal Malformation, Cardiac, Tracheo-esophageal, Renal, Limb anomalies.

### 3.3. Anorectal Manometry Characteristics

#### 3.3.1. Equipment

A range of approaches to anorectal manometry were reported. Fluid-perfused (water, saline or not described) catheters were used by the majority. Other reported methods included balloon or microballoon [16,37,49,50,69], solid-state [25], and microtransducer [40,45,64]. High-resolution manometry was utilized in six studies [22,34,52,54,64,72]. The number of sensors ranged from a single sensor region using the open tip method, to 256 sensors utilized to perform high-resolution anorectal manometry (HRAM) [52,64,72]. Although few studies described sensor spacing, the reported inter-sensor interval ranged from 0.5 mm to 2.5 cm [18,25,30,49,51,52,55,57,63,69,73]. Eight studies did not describe the methods used to perform anorectal manometry [20,23,28,33,39,70,71]. Anorectal manometry characteristics are summarized in Table 2.

#### 3.3.2. Preparation and Sedation

Few studies described the bowel preparation or sedation regimen utilized. With respect to preparation, the majority that described their approach reported the use of enemas, including glycerin and sodium phosphate. The bowel was not routinely prepared prior to anorectal manometry assessment in three studies [37,47,64]. A range of sedation strategies were reported, including chloral hydrate, ketamine, and nitrous oxide [26,28,29,30,31,34,35,38,39,47,48,49,54,57,63,69,73]. Anorectal manometry was performed without sedation in nine studies [14,22,32,43,50,52,55,57,67]. Reported approaches to preparation and sedation are summarized in Table 2.

**Table 2 jcm-12-02543-t002:** Summary of reported anorectal manometry specifications.

First Author	Year	Catheter Type	No. Sensors	Sensor Spacing	Preparation	Sedation/Anesthetic
Arnoldi [14]	2014	WP	4	-	Enema: performed evening prior	No sedation
Banasiuk [64]	2021	3D HRAM	256	-	No routine preparation. Saline enema, if required	-
Becmeur [23]	2001	-	-	-	-	-
Bhat [56]	2008	FP	Open tip	-	-	-
Burjonrappa [61]	2010	WP	-	-	-	-
Cahill [24]	1985	WP	8	-	-	-
Caldaro [43]	2012	WP	4	-	Enema: performed day prior	No sedation
Caruso [67]	2015	WP	4	-	-	No sedation
Caruso [72]	2021	3D HRAM	256	-	-	-
Chen [73]	1998	WP	4	0.5 cm	-	Rectal secobarbital: 6 mg/kg (<2 years)
Chung [22]	2018	WP HRAM	8	-	-	No sedation
Doolin [44]	1993	SaP	Open tip	-	-	-
El-Debeiky [25]	2009	S + WP	-	2.5 cm	-	-
Emblem [40]	1994	Micro-transducer	-	-	-	-
Emblem [45]	1997	Micro-transducer	-	-	-	-
Fukata [41]	1997	WP Foley catheter	1	-	-	-
Hedlund [46]	1992	WP	1	-	-	-
Heikenen [74]	1999	WP	8	-	-	-
Hettiarachchi [47]	2002	WP	4	-	No preparation	Ketamine
Huang [20]	2017	-	-	-	-	-
Husberg [26]	1992	WP ^11^	1	-	-	Ketamine or pentothal-N_2_0 (if required).
Ishihara [48]	1987	WP	1	-	Enema	Monosodium trichlorethyl phosphate (<4 years)
Iwai [15]	1988	FP Foley catheter	-	-	-	-
Iwai [68]	1993	FP	-	-	-	-
Iwai [65]	1997	FP	-	-	-	-
Iwai [66]	2007	FP	-	-	-	-
Keshtgar [69]	2007	Microballoon	4	1 cm	-	Ketamine
Keshtgar [49]	2008	Microballoon	4	1 cm	-	Ketamine
Kimura [27]	2010	WP Foley catheter	-	-	-	-
Kudou [28]	2005	-	-	-	-	Triclofos sodium
Kumar [57]	2010	SaP	4	1 cm	Glycerin enema	No sedation
Langemeijer [29]	1991	WP	Open tip	-	-	Nitrous oxide (<2 years)
Leung [70]	2006	-	-	-	-	-
Lin [30]	1996	WP	4	0.5 cm	-	Secobarbiturates (<2 years)
Lin [31]	2003	WP	3	-	-	Secobarbiturates; general anesthetic (<1 year).
Liu [32]	2004	WP	Open tip	-	-	No sedation
Martins [50]	1996	Balloon	2	-	-	No sedation
Mert [55]	2021	FP	8	0.5 mm	Enema: performed 2 h prior (70% sorbitol, glycerin, and tri-Sodium citrate hydrate)	No sedation
Mollard [75]	1991	FP	2	3 cm	-	-
Nagashima [62]	1992	WP	Open tip	-	Glycerin enema	-
Niedzielski [33]	2008	-	-	-	-	-
Ninan [71]	1994	-	-	-	-	-
Okada [21]	1993	WP	Open tip	-	-	-
Penninckx [16]	1986	Microballoon	1–3	-	-	-
Ray [76]	2004	FP	-	-	-	-
Ren [34]	2019	HRAM ^12^	-	-	-	Chloral hydrate
Rintala [35]	1990	SaP	1	-	-	Ketamine (if required)
Rintala [36]	1995	SaP ^13^	1	-	-	-
Rintala [58]	1993	SaP ^13^	1	-	-	-
Rintala [17]	1990	SaP ^14^	1	-	-	-
Rintala [59]	1995	SaP ^13^	1	-	-	-
Rintala [60]	1995	SaP ^13^	1	-	-	-
Ruttenstock [18]	2013	SaP	4	5 mm	Bowel washout day prior	-
Sangkhathat [63]	2004	WP	2	2 cm	-	Chloral hydrate
Schuster [19]	2000	WP	8	-	-	-
Schuster [42]	2001	WP	8	-	-	-
Senel [51]	2007	WP	4	0.5 cm	-	-
Sonnino [37]	1991	Double balloon	-	-	No preparation	-
Tang [52]	2017	WP HRAM	24	1 cm	Sodium phosphate rectal fleet enema	No sedation
Tong [38]	2011	WP	8	-	-	Chloral hydrate 1 mL/kg
Vital Junior [53]	2007	WP	4	-	-	-
Wang [54]	2016	HRAM	12	-	Enema	Chloral hydrate 0.3–0.5 mL/kg (if required)
Yang [39]	2009	-	-	-	-	Chloral hydrate 1 mL/kg

^11^ Manometry performed using either open-perfused catheter or water-filled cuff of a Portex endotracheal tube. ^12^ Catheter type not further defined (e.g., water-perfused, solid-state). ^13^ Methods as reported by Rintala; use of sedation unclear in this study population [35]. ^14^ Methods as reported by Rintala; use of sedation unclear in this study population [77]. FP: fluid-perfused; HRAM: high-resolution anorectal manometry; S: solid-state; SaP: saline-perfused; WP: water-perfused; -: not reported.

#### 3.3.3. Parameters

Assessment of a variety of anorectal manometry parameters was reported. The most commonly reported were resting pressure (anal, rectal or both) (55/63, 87%), rectoanal inhibitory reflex (RAIR) (52/63, 83%), and anal squeeze pressure (29/63, 46%). Reported parameters are summarized in Table 3. Criteria used to define assessed parameters were inconsistent; study definitions are summarized in Appendix A.

### 3.4. Anorectal Manometry Outcomes

Studies demonstrated significant variation with respect to the equipment, assessment protocols, and parameter definitions utilized. As such, absolute values for manometry outcomes were not combined. Key findings and study limitations are summarized in Table 4; absolute values of consistently reported parameters are summarized in Appendix A.

#### 3.4.1. Resting Pressure

The resting pressure was reported by 87% of studies. In most studies, this referred to the anal canal at rest [14,16,17,18,19,24,26,27,28,30,31,32,34,36,38,39,40,41,43,45,46,48,49,51,52,54,55,56,57,58,59,60,62,65,66,67,68,69,72,74], but two studies only reported rectal resting pressures [20,44], while another five reported both anal and rectal resting pressure [21,33,35,63,73]. In addition to assessment of the anorectum, one study also included assessment of the rectosigmoid region (resting pressure recorded 20, 15, 10, and 5 cm proximal to the anal verge) [62]. Six studies did not define the location of assessment [23,25,42,50,53,64,76].

Whilst the majority did not specify their approach to assessment of resting pressure [20,23,25,26,28,30,31,32,33,34,36,39,40,49,50,52,53,54,55,58,59,64,65,66,68,69,76], assorted approaches were described by those that did. Resting pressure was variably reported as the mean [14,24,41,43,46,51,56,63,67,72] or maximal [16,17,21,27,45,48,62] resting pressure, or according to study-specific measurements [18,19,35,38,42,57,60,74]. Manometry parameter definitions are summarized in Appendix A.

Reported absolute values of resting pressure varied between studies. Resting pressure ranged from 6.57 mmHg (measure of central tendency not described) in the incontinent group assessed by Martins et al. [50], to 75.75 ± 16.8 mmHg identified, using computerized vector manometry, by Schuster et al. [19]. Outcomes of anorectal manometry, including absolute values, are summarized in Appendix A.

Resting pressure was consistently reduced in children with anorectal malformations, compared with either normative values obtained from healthy children [42,46,57,64] or other diagnostic groups [72]. More complex malformations were typically associated with lower resting pressures than less complex malformations [23,33,43,45,48,49,51,62,76], although this was not uniform [63]. Reduced resting pressure was frequently identified in the setting of fecal incontinence [25,37,39,40,45,46,48,49,50,51,53,58,65,73,74,76]. With respect to obstructive symptoms, two studies identified significantly higher resting pressures in constipated than non-constipated patients [43,63], whilst another did not [58].

#### 3.4.2. Rectoanal Inhibitory Reflex

Assessment of the RAIR was reported in 52 studies (52/63, 83%); however, the criteria used to define the RAIR varied [14,15,16,17,18,19,20,21,22,23,24,25,26,27,28,29,30,31,32,33,34,35,36,37,38,39,41,43,44,45,46,47,49,50,51,53,54,55,57,58,59,60,61,63,64,66,68,69,73,74,75,76]. For example, a positive RAIR was defined according to an absolute pressure decrease (i.e., >5 mmHg, >15 mmHg, >1 kPa) [14,49,57,74]; percentage reduction relative to resting pressure (i.e., >25%, >50%) [64,73]; or qualitative criteria (i.e., “relaxation of the internal sphincter”, “decrease of anal pressure”, “anal response”, “clear pressure decrease”) [24,26,44,46,50]. In addition, there were variable inclusion of criteria related to duration of relaxation (i.e., “sustained for 5 seconds”) [22], rectal balloon volume (i.e., [reflex present at] 60 mL, 100–150 mL) [47,69] and/or reflex reproducibility (i.e., “present on 3 consecutive attempts”) [36,58]. The proportion of the cohort with an RAIR identified varied maximally, ranging from 0% [35,50,60,65,66,68,76] to 100% [16,18,24,25,34,40,55,59].

#### 3.4.3. Squeeze Pressure

Squeeze pressure was assessed by 29/63 (46.0%) studies [15,22,25,29,32,37,38,40,42,43,44,45,46,50,52,53,55,60,63,64,65,66,67,68,70,71,72,74,76]. Variable criteria were used to define and assess squeeze pressure, including average/maximal anal squeeze pressure (expressed as absolute values or increment relative to resting pressure), voluntary contraction pressure, voluntary sphincter contraction, peak squeeze pressure, maximal mean segmental pressure during squeeze, and squeeze pressure of the high-pressure zone. Reported squeeze pressures ranged from 0 mmHg (prior to gracilis transposition for treatment of severe fecal incontinence) [37] to a median of 121.7 mmHg (range 38.2–46.8 mmHg) identified by Banasiuk et al. [64]. Of those that reported comparison of squeeze pressure by complexity of malformation type, there was some evidence that more complex malformation types had reduced squeeze pressure relative to less complex types [55,62]; however, this was not uniform [15].

#### 3.4.4. Normal Findings

In order to define abnormalities of anorectal function, an understanding of “normal” pediatric physiology is needed. However, the definitions of what was considered normal were either not reported or incompletely described by most studies included in this review. For example, there was variability with respect to both anorectal manometry parameters considered to be significant, and the absolute pressure values within parameters that were considered within normal range. Consequently, the understanding of “normal” varied markedly between studies (Appendix A).

#### 3.4.5. Post-Operative Outcome Assessment

Established measures were used to assess post-operative bowel function in 36 studies [14,15,18,20,21,22,23,25,27,28,32,33,34,38,39,40,41,42,43,45,47,49,51,55,56,60,61,64,65,66,67,68,69,70,72,76]. Measures included the Kelly, Rintala, Wingfield, and Krickenbeck continence scoring methods, with the Kelly score and its variants being utilized most frequently [15,21,25,28,33,38,39,41,42,51,52,56,65,68]. Study-specific assessments of bowel function were reported by 19 studies. Bowel function was either not assessed or not described by the remainder. Few studies provided symptom definitions. In those that reported the definitions utilized, inconsistent criteria were demonstrated. For example, constipation was variably defined by frequency (e.g., “<1 bowel action per day”, “bowel action every 2–3 days”), management requirements (e.g., “cisapride or laxatives but without enemas”, “enema required daily to achieve bowel action”, “dietary requirements”), and/or subjective impression of severity (e.g., “severe, unmanageable”) [33,35,37,58,62,73]. Post-operative bowel function assessment measures and findings are summarized in Table 5.

## 4. Discussion

A significant proportion of children with anorectal malformations experience disorders of evacuation and/or fecal incontinence following operative repair. Continence may be affected by a variety of factors, including malformation type, associated sacro-/spinal anomalies, and operative repair type, including intra- and/or post-operative complications. Due to the variable etiology of fecal incontinence and evacuation disorders, symptoms alone are often insufficient to direct treatment in cases refractory to conservative management [6]. Anorectal manometry may be used to investigate the pathophysiology underlying anorectal dysfunction. To our knowledge, this review is the first to systematically evaluate post-operative anorectal manometry performed in children with anorectal malformations.

In assessing 63 studies, our overall finding was a complete lack of consistency between manometry protocols, analysis of data, and interpretation of findings. This echoes the conclusion of a similar review of anorectal manometry performed in adult populations [7]. Despite studies identifying abnormalities in anorectal function, definitions of normal were rarely provided or vaguely described. Collectively, this makes interpretation of the findings difficult, whilst comparison of data between studies is impossible.

### 4.1. Manometry Outcomes

In 1834, Roux de Brignoles described the importance of preserving the fibers of the sphincter mechanism during anorectal malformation repair, demonstrating the long-standing recognition of their importance [2]. In contemporary practice, anorectal manometry may be used to assess post-operative function of the sphincter complex: activity of the internal anal sphincter (IAS) is understood to provide the majority of resting anal pressure, whilst the external anal sphincter (EAS) is largely responsible for voluntary contraction (squeeze pressure). Hypotonia may, therefore, be associated with presentations of fecal incontinence, whilst increased resting pressure may underlie some rectal evacuation disorders.

The most consistently reported findings identified by studies included in this review were of reduced resting pressure in children with repaired anorectal malformations (usually relative to unspecified normative values or other diagnostic cohorts), with the decrease particularly evident in more complex malformation types and/or in the setting of non-retentive fecal incontinence [23,25,33,37,39,40,42,43,45,46,48,49,50,51,53,57,58,62,64,65,72,73,74,76]. Following re-operation for the management of severe fecal incontinence, improvements in resting pressure were similarly associated with improved continence [21,37].

These findings are not unexpected, considering the underlying developmental abnormalities and the subsequent impacts of operative intervention on an underdeveloped continence apparatus in these children. From a clinical perspective, the thresholds at which fecal incontinence may be attributed to sphincter dysfunction are more difficult to determine. Despite reduced resting pressure, not all children with anorectal malformations experience incontinence. Poor concordance between severity of fecal incontinence and tests of anorectal function have been similarly demonstrated in adult populations [77,78]. Whilst these findings may appear to limit the utility of anorectal manometry, several factors may contribute to this apparent discrepancy. Firstly, given the wide range of resting pressures reported by studies included in this review, and a lack of “normal” values, identifying a precise threshold at which incontinence may be expected is challenging. The need to establish optimal manometric measurements for the diagnosis of anorectal dysfunction has been emphasized for this technique globally [6], and the additional challenges inherent to the pediatric setting are well-recognized [9].

Secondly, continence may be impacted upon by factors extrinsic to the anorectum and thus not evaluated by anorectal manometry. The regulation of defecation and its control (continence) is multifactorial; it is reliant on the interplay between key anatomical structures (principally the colon, anorectum, and pelvic floor musculature) and physiological systems (principally nervous, muscular, hormonal, and cognitive) [79]. As colonic motor activity propels luminal contents distally, progressive rectal distension produces the defecatory urge. If the timing is unsuitable, voluntary contraction of the EAS results in deferral of defecation, and retrograde colonic motor patterns return luminal contents to the sigmoid colon [79]. Alternatively, the expulsive phase sees reversal of the rectoanal pressure gradient through (1) voluntary relaxation of the EAS; (2) reflex relaxation of the IAS and pelvic floor musculature; and (3) reduction of the anorectal angle. Following evacuation, the basal rectoanal pressure gradient is restored and continence is re-established [79].

Given the complexity of interactions required for successful control of defecation, it is perhaps unsurprising that a significant proportion of children with anorectal malformations experience disorders of defecation, despite careful anatomic reconstruction [5,80,81]. Characteristic congenital defects of the anus and rectum may be intuitively associated with impaired continence. In addition, under-developed pelvic musculature, surgical interventions (particularly in the setting of revision procedures), and associated sacro-/spinal anomalies may impact upon the neuromuscular integrity of the continence system [1,82,83].

Prior to defecation, rectal filling elicits the RAIR: reflex relaxation of the IAS facilitates anal mucosal “sampling” and discrimination between solid, liquid, and gaseous luminal content [84,85]. Several studies described a reduced or absent RAIR in children with anorectal malformations [57], including in association with high [versus low] malformations [57] and/or neurospinal dysraphism [14]. Others described better continence outcomes associated with presence of an RAIR, including higher scores on assessments of bowel function [14,31], lower prevalence of fecal incontinence [14,26,29,46,53], and/or higher anal resting pressure [26,46,59].

In this review, the array of approaches used to define and assess the RAIR made it challenging to understand the contribution of the reflex to the continence outcomes. It is likely that, alongside any true differences in prevalence, the variability observed was significantly influenced by the inconsistent criteria used. Despite these limitations, there is some evidence that presence of the RAIR impacts positively upon bowel function in this cohort, and, in concert with other parameters, may aid efforts to prognosticate continence outcomes.

As emphasized by Kumar et al., in the setting of anorectal malformations, RAIR absence is often described in relation to IAS dysfunction (resulting from congenital abnormalities and/or surgical disruption) [57]. However, the higher rectal balloon volumes required to elicit the RAIR in their studied cohort, and reflex absence in children with severe sacral anomalies, led the authors to emphasize the other arm of the reflex arc: the impact of disrupted sensory perception (rather than solely IAS dysfunction) in these patients [57]. Future work should seek to carefully characterize the relationship between presence of an RAIR, key medical characteristics (including malformation type, operative repair approach, and sacrospinal anomalies), bowel function, and response to interventions. This may facilitate understanding of the contribution of the RAIR, and utility in relation to prognostication, within the broader continence apparatus.

Proximal to the anal canal, there is increasing recognition of role of the colon in maintaining fecal continence, particularly through the regulation of rectal filling [86,87,88,89]. Included in this review, Heikenen et al. evaluated both colonic and anorectal motility in children with fecal incontinence following anorectal malformation repair [74]. Whilst a considerable proportion of the children demonstrated reduced resting pressure (60%), propagation of an “excessive” number of colonic motor complexes into the neorectum was demonstrated [74]. Although subject to significant limitations, the findings suggest the refractory fecal incontinence demonstrated by their cohort resulted from multiple factors, including those arising beyond the limits of the anorectum [74]. Our current understanding of colonic motility in this cohort remains poor; however, such studies may be of value in helping to understand the ongoing symptoms in children after repair of anorectal malformations.

While not assessed in this paper, several included studies used other modalities, in addition to anorectal manometry, to help define anorectal abnormalities [41,43,47,49,69]. For example, correlation between findings of manometry and anal endosonography were demonstrated in two studies [49,69]. However, these findings have been questioned, given the limited size and composition of the cohort and the probable technical limitations introduced by the use of a Foley catheter for anorectal manometry [41]. Caldaro et al. similarly utilized manometry and anal endosonography in combination to prognosticate response to treatment [43]. Symptoms in the setting of IAS disruption (identified using anal endosonography) were found to be responsive to biofeedback and laxatives if anal resting pressure (identified using manometry) was greater than 20 mmHg [43]. Utilization of a combination of select investigations is recommended to understand structure and function of the anorectum [6]. The need to establish classification systems encompassing outcomes of multiple tests of anorectal structure and function has been highlighted previously, and would similarly be of benefit to this patient cohort [6].

### 4.2. Manometry Outcomes and Clinical Correlates

The approach to bowel function assessment was highly variable between studies. Few studies provided symptom definitions, designating groups with constipation or fecal incontinence without specifying the diagnostic criteria. Those that did provide an explanation of their diagnostic terminology demonstrated inconsistencies. For example, constipation was defined as “less than three bowel actions per week”, “less than one bowel action per day”, and “enema required daily to achieve bowel action” [32,62,63]. Similarly, criteria used to report severity often lacked specificity. Studies often assigned grades or terminology (such as minor or infrequent), without specifying the features of each category. This discordant approach to outcome assessment limits meaningful comparison of findings between studies, identification of manometry-symptom correlates, and response to manometry-guided management strategies.

### 4.3. Practice Variability

Variability has been a common theme throughout this review. We have identified variability in cohort reporting, bowel function assessment, and symptom profiles. Variability in practice has been highlighted, with notable differences in the equipment, protocols, motility criteria, and interpretation used by included studies. Despite the impact of the manometry catheter and assessment protocol on absolute values achieved, notably few studies adequately described the catheter and approach utilized, whilst seven studies provided no description of their approach [20,23,28,33,39,70,71]. This presented a significant challenge when attempting to compare outcomes.

Fundamental to improving the consistency of this work is a coordinated effort to standardize anorectal manometry assessment, its interpretation, and reporting of findings, as has recently been developed for adult anorectal manometry studies [6,7]. Future studies should utilize a recognized manometry protocol and reporting framework; consensus statements have been developed for this purpose [8,9]. Similarly, this must be accompanied by robust reporting of relevant cohort medical characteristics (particularly malformation type, approach to operative repair, and associated anomalies affecting the spine and sacrum); evaluation of bowel function; and assessment of their relationship to manometry findings (Figure 2). Development of minimum standards should be considered, to guide reporting of key cohort characteristics in this population.

## 5. Limitations and Conclusions

This review was limited to children following anorectal malformation repair. Studies including manometry outcomes in this cohort may have been excluded if the findings were not separated by age and/or diagnostic group. Consequently, the included studies may not reflect all anorectal manometry findings in this cohort. Similarly, our review was restricted to studies published in English and may be subject to a language bias.

Whilst altered anorectal function may be intuitively presumed to impact upon continence, regulation of defecation is multifactorial. The prevailing limitation of our current understanding of bowel dysfunction following anorectal malformation repair is the failure to place manometry findings into this wider context. Along with standardization of the approach used to perform anorectal manometry, this should be the focus of future work assessing anorectal function in this cohort. To support this process, the development of reporting guidelines for cohort characteristics and clinical outcomes should be considered, specific to children with anorectal malformations undergoing motility assessment. Whilst high-resolution techniques may provide greater insight into anorectal structure and function, interpreting manometry findings within the context of the broader continence mechanism is essential to enhancing our understanding of the long-term bowel dysfunction experienced by this cohort.

## Figures and Tables

**Figure 1 jcm-12-02543-f001:**
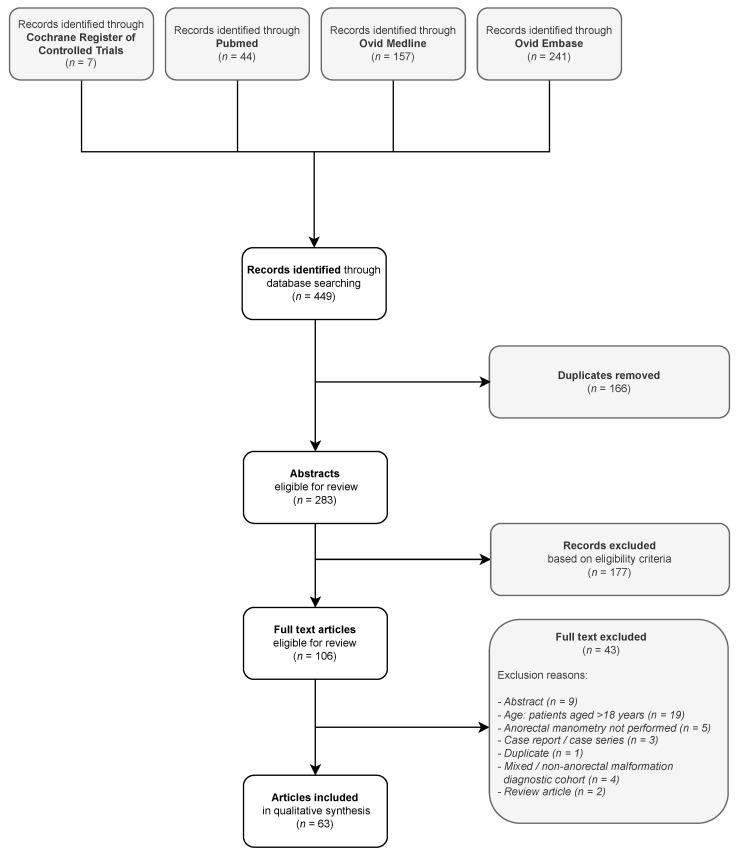
Study selection flow diagram.

**Figure 2 jcm-12-02543-f002:**
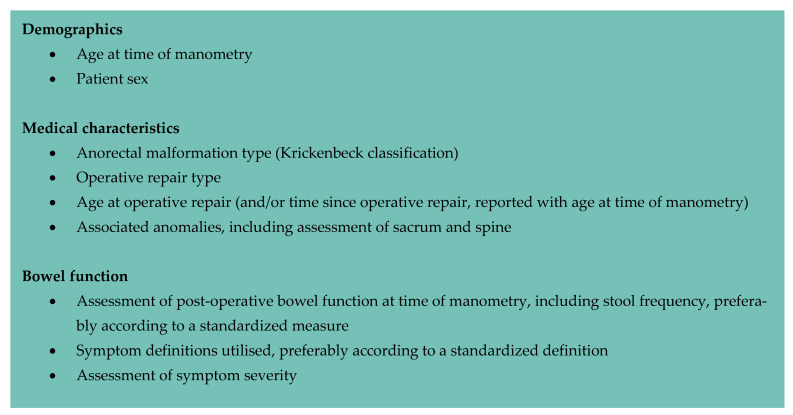
Suggested key cohort medical characteristics for reporting anorectal manometry studies performed in children with a repaired anorectal malformation.

**Table 3 jcm-12-02543-t003:** Reported anorectal manometry parameters. Comparable parameters are grouped.

First Author	Resting Pressure	Squeeze Pressure	Endurance Squeeze	Anal Canal Length	High Pressure Zone (HPZ)	Sensation	Maximum Volume tolerated	RAIR	Cough	Rectal Compliance	Other
Arnoldi [14]	●					●		●			
Banasiuk [64]	●	●			●	●		●			Resting and squeeze pressures of the puborectalis muscle ^16^
Becmeur [23]	●							●			
Bhat [56]	●										
Burjonrappa [61]						●		●			
Cahill [24]	●							●			
Caldaro [43]	●	●				●	●	●			
Caruso [67]	●	●									Sphincter symmetry
Caruso [72]	●	●									Sphincterial asymmetry ^17^, rectal sensitivity ^18^
Chen [73]	● ^19^							●			Anorectal pressure gradient
Chung [22]	●	●				●		●			
Doolin [44]	●	● ^20^						●			Spontaneous contraction pressure; rectal contraction frequency (number/minute)
El-Debeiky [25]	●	●			●	●	●	●			
Emblem [40]	●	●						●			
Emblem [45]	●	●									
Fukata [41]	●							●			
Hedlund [46]	●	●						●			Average rectal volume
Heikenen [74]	●	●						●			
Hettiarachchi [47]								●			Manometric rectal score, IAS length and activity scores
Huang [20]	● ^21^							●			Active systolic blood pressure; rectal compliance ^22^
Husberg [26]	●							●			
Ishihara [48]	● ^23^										
Iwai [15]		●									
Iwai [68]	●	●				●	●	●		●	Anorectal pressure difference
Iwai [65]	●	●	●			●	●	●		●	
Iwai [66]	●	●		●		●	●	●		●	
Keshtgar [69]	●							●			
Keshtgar [49]	●							●			
Kimura [27]	●							●			
Kudou [28]	●				● ^24^			●			
Kumar [57]	●				●			●			
Langemeijer [29]		●				●		●			
Leung [70]		●									
Lin [30]	●							●			Anorectal pressure profile, slow wave activity of the anal canal.
Lin [31]	●							●			
Liu [32]	●	●						●			
Martins [50]	●	●						●	●		Duration of sustained voluntary contraction, perianal stimulation, crying, pressure curve during balloon withdrawal.
Mert [55]	●	●		●		●	●	●			First urge to defecate volume; area under curve during maximum voluntary squeeze.
Mollard [75]						●		●		●	Maximal anal resting closure pressure: maximal anal pressure minus rectal pressure.
Nagashima [62]	●					●	●			●	Anorectal pressure difference. ^25^
Niedzielski [33]	●			●				●			Rectoanal pressure gradient, reflex pressure amplitude.
Ninan [71]		●									
Okada [21]	● ^26^							● ^27^			
Penninckx [16]	●							●			Anal slow pressure waves at rest
Ray [76]	●	●			●			●			
Ren [34]	●				●			●			
Rintala [35]	●							●			Slow pressure-wave activity of the anal canal
Rintala [36]	●							●			
Rintala [58]	●							●			
Rintala [17]	●							●			Slow pressure-wave activity of the anal canal
Rintala [59]	●							●			
Rintala [60]	●	●						●			
Ruttenstock [18]	●			●				●			
Sangkhathat [63]	●	●						●			Resting rectoanal pressure gradient ^28^
Schuster [19]	● ^29^			●	●			●			Segmental and total asymmetry indexes; vector volume ^30^
Schuster [42]	● ^31^	● ^32^		● ^33^	●						Sphincter length; vector volume ^34^
Senel [51]	●							●			
Sonnino [37]		●				●		●			
Tang [52]	●	●									Inter-quadrant pressure asymmetry index ^35^
Tong [38]	●	●			●			●			Asymmetry index, vector volume ^36^
Vital Junior [53]	●	● ^37^		●				●	●		Maximum pressure, pressure during perianal stimulation, pressure asymmetry
Wang [54]	●							●			
Yang [39]	●				●			●			
Prevalence	55	29	1	7	10	14	7	52	2	5	

^16^ Resting and squeeze pressures of the puborectalis muscle: recorded in segments covering its anatomical location. ^17^ Sphincterial asymmetry: difference of resting and squeeze pressure >20% between four cardinal anal segments on 3D analysis. ^18^ Volume (mL) of rectal balloon inflation which elicited defecatory urge. ^19^ Resting anal pressure, resting rectal pressure, and anorectal pressure gradient. ^20^ Voluntary contraction pressure. ^21^ Resting rectal pressure. ^22^ It is unclear how anorectal manometry was used to facilitate assessment of active systolic blood pressure; calculation of rectal compliance was not defined. ^23^ Maximum anal canal static pressure. ^24^ LFS: length of internal functional sphincter. ^25^ Anorectal pressure difference: difference between maximum anal pressure (at anal verge) and resting rectal pressure. ^26^ Rectal pressure and maximum anal canal pressure. ^27^ Anorectal reflex: not further defined. ^28^ Resting rectoanal pressure gradient: the difference between the resting rectal pressure and the resting anal pressure. ^29^ Maximal mean segmental pressure at rest. ^30^ Vector volume: all sampled pressure values (as vectors) and anal canal length were combined, and the volume of the imaginary pressure cylinder was calculated. ^31^ Maximal mean segmental pressure at rest. ^32^ Maximal mean segmental pressure during squeeze. ^33^ Functional anal canal length: the distance between the anal verge and the proximal functional border of the anal canal, represented by the increased pressure of the muscular tube, compared to the baseline pressure in the rectum. ^34^ Vector volume: calculated volume of the imaginary pressure cylinder. ^35^ Inter-quadrant pressure asymmetry index (∆*p*): inter-quadrant pressure difference divided by the maximal quadrant pressure, expressed as a percentage. ^36^ Vector volume: direct representation of individual force of the anorectal contractile units; integration of pressure and length. ^37^ Voluntary contraction pressure. IAS: internal anal sphincter; LFS: length of internal functional sphincter; RAIR: rectoanal inhibitory reflex.

**Table 4 jcm-12-02543-t004:** Summary of key anorectal manometry findings.

First Author	Manometry Population Summary	Summary of Key Anorectal Manometry Findings	Key Limitations
Arnoldi [14]	Toilet-trained children with anorectal malformations with a good predicted prognosis ^38^	Significantly lower resting pressure (RP) in children with neurospinal dysraphism (ND) and/or children with a neonatal colostomy, than children without; RP in patients without ND or neonatal colostomy comparable to control children. In patients with a pathologic Rintala score, RP was significantly lower and RAIR was identified in less than half; the reflex was identified in 100% of children with a normal Rintala score.	Exclusion based on toilet training status introduces potential selection bias (excluding children who may never attain continence); although the homogenous population is a strength, findings not generalizable to the wider anorectal malformation cohort.
Banasiuk [64]	Children who had undergone surgery for anorectal disorders, including anal atresia	Lowest RP, squeeze pressure (SP), and pressure of the puborectalis muscle were observed in the anal atresia group; these parameters were significantly lower than healthy controls. Patients with non-retentive fecal incontinence demonstrated significantly decreased RP compared to healthy controls.	Clinical characteristics, such as gender of diagnostic subgroups and operative repair type unknown; small, heterogenous anorectal malformation cohort: wide age range with diverse malformation types.
Becmeur [23]	Children following three-flap anoplasty for primary or re-do repair	Normal anorectal manometry findings (RP, RAIR) in children with low or intermediate imperforate anus. High anorectal or complex caudal malformations associated with poor anorectal manometry findings and clinical score.	Small, heterogeneous cohort; significant proportion of cohort with comorbidities likely to impact continence; very limited description of manometry methods/results.
Bhat [56]	High or intermediate anorectal malformation, following sigmoid colostomy formation but prior to PSARP	Pre-PSARP mean rectal pouch pressure reported to be similar to the mean post-PSARP and post-colostomy closure anal canal pressures. Kelly’s score (functional assessment) reported to be related to anal canal pressure. The authors reported that these associations demonstrated the ability to predict post-operative continence pre-operatively.	Small cohort; associated anomalies not reported; limited description of statistical methods to support findings (e.g., correlation between anal canal pressures and Kelly score); short follow up period to support conclusion that post-operative continence may be predicted pre-operatively (median age 29 months (range 19–60) at assessment).
Burjonrappa [61]	Patients with megarectum following surgery for anorectal malformation	All children who underwent anorectal manometry prior to excision of megarectum (*n =* 5) had an intact RAIR. Two children underwent colonic manometry which demonstrated “hyperperistalsis” and “a neorectum very sensitive to distension”.	Small, heterogeneous manometry cohort (six children); select population (children with megarectum); limited manometric assessment with variable reporting of findings.
Cahill [24]	Patients with anorectal malformation following PSARP	Four of five patients had “normal” rectal sensation and mean anal canal pressures following PSARP, suggesting the procedure is applicable to young infants with high imperforate anus.	Small cohort; limited description of manometry techniques, findings, and interpretation; too young at follow up to adequately assess bowel function to support the interpretation of findings.
Caldaro [43]	Neurologically healthy children, >4 years, with constipation/FI, following anorectal malformation repair	Average anal resting pressure (aARP) significantly higher in the low malformation group, than intermediate or high malformation groups. Fecal incontinence in the setting of IAS disruption (identified using 3D endoanal ultrasound) responded to biofeedback +/− laxatives if aARP > 20 mHg, whereas daily enemas were necessary if aARP < 20 mmHg. Statistical correlation identified between manometric, endosonographic, and clinical findings; useful to define most appropriate treatments based on anal sphincter assessment and understanding of continence “potential”.	Small malformation subgroups limit strength of findings with respect to malformation types.
Caruso [67]	Neurologically heathy children >4 years with “true” FI following anorectal malformation repair	Anorectal manometry can evaluate potential sphincteric recovery after biofeedback for the treatment of FI; further prognostic benefit if correlated to morphologic evaluation with MRI. Alternative treatment should be considered in patients with unfavorable pre-treatment assessment. Improvement in manometric values associated with improved clinical score after biofeedback therapy.	Manometry/MRI assessment would benefit from clarification of scoring; relatively small subgroups (determined by pre-operative assessment).
Caruso [72]	Children with FI or bowel dysfunction not responsive to conventional laxative treatment, receiving transanal irrigation	Sphincteric anomalies observed more frequently in anorectal malformation group (compared to Hirschsprung disease [HD]; neurological impairment [NI]; functional fecal incontinence [FFI] groups), with asymmetry and lower RP. Sphincter anomalies, identified using 3D HRAM, were the most important prognostic factor for TAI efficacy: associated with worse scores of function, and slower improvement following TAI initiation.	Small anorectal malformation cohort; surgical repair type not reported.
Chen [73]	All children with anorectal malformation repaired by a single surgeon	Resting rectal pressure (RrP) lower and anorectal pressure gradient (ARPG) higher in the constipated than the non-constipated children. The RrP was higher and the resting anal pressure and ARPG lower in the patients with soiling.	Limited cohort data provided for manometry cohort; limited by technology available; idiosyncratic symptom assessments may limit comparability.
Chung [22]	Toilet-trained children following PSARP or LAARP without neurological comorbidities or cloacal malformation	The majority of patients demonstrated normal sphincteric resting pressure following LAARP. Patients demonstrated higher sphincteric resting pressure and bowel function scores following LAARP (compared with PSARP).	Small operative/malformation subgroups; difference in time period between operative subgroups—results may be confounded by evolution of pre- and post-operative care; PSARP subgroup older than LAARP cohort and includes subjects >18 years, limiting comparability.
Doolin [44]	Children following repair of anal atresia	Manometry findings were not significantly correlated with functional outcomes. No objective criteria were identified that could evaluate the patient’s clinical result or guide therapy.	Malformation classification not known for all patients; limited by technology available.
El-Debeiky [25]	Males with high anorectal malformation treated with laparoscopic-assisted pull-through	“High” resting pressure that decreased on straining in patients without soiling (*n =* 7) and “low” in two patients with soiling.	Small cohort; unclear age at assessment limited, qualitative reporting of manometry findings, without clarification of interpretation (e.g., “high” versus “low” resting pressure); assessment of correlation between manometry findings and function not reported.
Emblem [40]	Adolescents with low anorectal malformations following repair	Graded continence assessment was significantly correlated with anal canal resting and squeeze pressure. Resting and squeeze pressure demonstrated significant sex differences: males had significantly better functional results and higher anal pressures.	Small cohort; limited by technology available; limited information provided regarding technique.
Emblem [45]	Adolescents with anorectal malformations following repair	Continence correlated with resting and squeeze pressures of the anal canal; resting pressure was the strongest predictor of continence. Resting pressure correlated with diagnosis, being highest in the control group, intermediate in the low anorectal malformation group, and lowest in the intermediate/high malformation group. Squeeze but not resting pressure was sex dependent.	Small malformation subgroups limit strength of findings with respect to malformation types; cohort details limited (type of malformation and surgical repair); limited by technology available; potentially subject to non-responder bias.
Fukata [41]	Patients with high or intermediate anorectal malformations	Presence of IAS on endoanal ultrasound associated with improved Kelly score but not correlated with manometry findings. Patients with high anomalies had congenitally rudimentary EAS as demonstrated on anal endosonography.	Small cohort; limited by manometry techniques available; functional outcome reported as Kelly score without description of symptom profile; methods/outcomes to assess statistical correlations not reported.
Hedlund [46]	Patients with anorectal malformations following PSARP, without major sacral malformation	Anal resting tone (ART) and anal squeezing pressure (ASP) subnormal in most patients, with soiling more common in patients with very low ART (<40 cm H_2_O) and a low ASP (<100 cm H_2_O). Constipation more common in patients with a large rectal volume. Presence of a rectoanal inhibitory reflex correlated to both a comparatively high ART and low incidence of soiling.	Statistical assessment of correlation between manometry findings and clinical outcomes not reported; limited reporting of patient cohort data; limited reporting of manometry outcomes.
Heikenen [74]	Children with FI refractory to standard medical therapy following repair of anorectal malformation	Anal RP was reduced in 60% of children, all with refractory fecal incontinence; average resting pressure 19.5 mmHg.	Limited reporting of anorectal manometry technique and outcomes; small cohort; anorectal manometry not performed in all children.
Hettiarachchi [47]	Children with chronic constipation and/or FI following anorectal malformation repair	Manometric IAS scores correlated with functional scores (assessed using the modified Wingfield score (MWS)), as did overall manometric scores (IAS + rectal score). Combined manometric and MRI scores showed a correlation with MWS; however, MRI scores alone did not.	Small, heterogenous cohort, particularly with respect to age, repair, malformation type, associated anomalies, and functional outcomes; idiosyncratic scoring system limits comparability; limited objective reporting of manometry findings.
Huang [20]	Female patients with rectovestibular fistula	No significant difference in resting rectal pressure between surgical groups, despite lower rates of FI and constipation in the modified-PSARP cohort; however, assessment of manometry findings and correlation with symptom subtypes does not appear to have been performed.	Manometric assessment of “active systolic blood pressure” and measurement of rectal compliance unclear; assessment of correlation between symptom groups and manometry findings not performed.
Husberg [26]	Children with high or intermediate anorectal malformation	Patients with demonstrated rectoanal inhibitory reflex (RAIR; 32/43) demonstrated significantly better anal continence and anal pressure. All patients with absent RAIR demonstrated constant soiling; patients with positive RAIR had significantly better continence.	Limited by technology and surgical repair techniques available; limited reporting of manometry findings; bi-national cohort, assessed utilizing different anorectal manometry techniques; small sub-cohort > 4 years contributing to functional evaluation.
Ishihara [48]	Patients with anorectal malformations following repair	Patients with normal defecation or constipation demonstrated higher maximum anal canal static pressure, compared with those with FI. Maximum anal canal static pressure was 50% or more in patients with translevator type malformation (normal defecation), 30–50% in those with supralevator type with normal defecation or constipation, and 30% or less in those with supralevator type with FI.	Limited by technology available; limited reporting of cohort demographic and medical details; few comparable manometry findings (results expressed as percent maximum static pressure of control group).
Iwai [65]	Children undergoing biofeedback training for FI following anorectal malformation repair	Anal resting pressure significantly lower and anorectal reflex absent in children with FI following anorectal malformation repair, compared with control group (encopresis). Anal resting pressures not affected by biofeedback therapy; however, voluntary sphincter function and rectal sensation improved. Congenital absence/weakness of IAS in children with anorectal malformations reported to prevent attainment of continence, even with improvement in EAS function/rectal sensation following biofeedback therapy.	Small cohort; limited cohort characteristics provided, including median cohort age at assessment, operative repair type, or characterization of FI; limited description of manometry technique.
Iwai [66]	Children with severe constipation following anorectal malformation repair, treated with herbal medication	Clinical scores improved following treatment with Dai-Kenchu-To Threshold sensation volume, maximum tolerable volume, and rectal compliance decreased post-treatment, suggesting regular stooling promoted by Dai-Kenchu-To led to secondary improvements in rectal reservoir function.	Very small anorectal malformation sub-cohort; limited description of manometry technique; statistical comparison of manometry findings in anorectal malformation cohort not performed, due to small cohort size.
Iwai [68]	Children undergoing biofeedback training for FI following anorectal malformation repair	Biofeedback improved voluntary sphincter function; anal resting pressure and ARPD were not significantly affected. Children who had a good response (3/8 demonstrated improved Kelly score) had adequate anal resting pressure both before and after biofeedback.	Small cohort; responders had three or more sessions (non-responders only one or two sessions), potentially confounding this result; limited description of manometry techniques.
Iwai [15]	Patients with anorectal malformations following repair	Voluntary contraction pressure (VCP) did not differ significantly between anomaly types (low, intermediate, high); however, it was significantly lower in children with intermediate and high anomalies, than controls. Patients with high anomalies and Kelly score < 2 (*n =* 3) had VCP < 20 cmH_2_O (14.7 mmHg). Suggested that patients with high anomalies may develop voluntary continence if EAS developed through bowel training.	Manometry not performed in all patients; limited description of cohort characteristic; limited manometric assessment performed; statistical assessment of correlation between manometry findings and bowel function scores not assessed/reported.
Keshtgar [69]	Children undergoing excision of megarectum for intractable FI	Pre-operatively (prior to excision of megarectum), there was a significant relationship between functional assessment of the IAS on manometry and morphologic evaluation on anal endosonography.	Heterogenous cohort with respect to malformation and primary repair types; quantitative manometry results not provided.
Keshtgar [49]	Children with intractable FI/constipation following anorectal malformation repair	Resting sphincter pressure in children with low malformations was comparable to reference groups (chronic constipation, healthy children) Children with high anomalies had significantly lower resting pressure and poorer fecal continence; however, this group was excluded due to age. Anorectal manometry and anal endosonography correlated	Children with high malformations excluded due to inclusion of patients >18 years of age; heterogenous cohort with respect to malformation and primary repair types; idiosyncratic classification of malformations (high or low)
Kimura [27]	Infants managed for high anorectal malformation	No significant difference in maximal anal resting pressure between laparoscopic and open anorectoplasty cohorts. Trend toward positive anorectal reflex in laparoscopic group; however, the difference was not significant. Outcomes with respect to anorectal function after open anorectoplasty considered to be non-inferior to laparoscopic approach.	Limited by technology used; minimal description of manometry techniques and interpretation; 17-year study period introduces potential confounding factors related to other treatment advances; shorter mean follow up following laparoscopic procedure (11.8 versus 4.3 years).
Kudou [28]	Children following LAARP or PSARP for management of high anorectal malformation	Clinical score, RP, and length of internal functional sphincter were comparable between LAARP and PSARP groups. A rectoanal relaxation reflex (RAR) was identified in 62% of the LAARP group and 29% of the PSARP group; however, the difference was not significant.	Small cohort; LAARP group significantly younger at evaluation.
Kumar [57]	Children with anorectal malformations following repair	Children with repaired anorectal malformations demonstrated shorter anal canal length, reduced RP, and impaired RAIR in comparison to healthy, age-matched controls.	Functional outcomes not reported/assessed in comparison to manometric findings; relatively small malformation/age subgroups.
Langemeijer [29]	Patients with high malformation following PSARP	Inhibitory reflex identified in one of 39 patients; identified in the only continent patient in the series. No discernible functional difference following “new” PSARP in comparison to older repair techniques.	Limited by technology available; limited description of manometry technique and interpretation; limited manometric assessment performed; statistical assessment of correlation between manometry findings/function not performed/reported.
Leung [70]	Children >5 years old with FI following anorectal malformation repair	There was a significant increase in mean anal sphincter squeeze pressure, associated with a significant improvement in fecal continence, following electrical stimulation and biofeedback exercise of pelvic floor muscles	Small cohort; manometry not performed in all patients; limited manometric assessment performed, with limited description of equipment, technique, and interpretation.
Lin [30]	Children with high or intermediate malformation following PSARP or R-APSA	No significant difference in resting anal pressure between surgical or age groups. Prevalence of RAR differed neither between PSARP with and without an internal sphincter-saving procedure, nor PSARP and R-APSA groups. The IAS-like structure not essential for continence or RAR: in the presence of a damaged IAS-like structure, or its complete absence, adaptation of the neoanus through reinnervation of the bowel end was speculated to cause it to behave like an IAS, accounting for the appearance of RAR in patients without IAS-saving procedures.	Small cohort in the context of multiple operative subgroups; study-specific assessment of bowel function may limit comparability of findings.
Lin [31]	Patients with high or intermediate malformations following LAR or PSARP	Positive RAR identified in significantly greater proportion of patients post LAR compared to PSARP; presence of RAR significantly correlated with “acceptable” number of bowel actions per day (1–4) in both groups. Significantly earlier recurrence of RAR identified in LAR group. Comparable resting sphincteric pressure; resting rectal pressure significantly lower in LAR group. Early post-operative manometry findings more favorable following LAR.	Small operative sub-groups; assessment of bowel function based on stool frequency, without report of associated symptoms (e.g., constipation, FI).
Liu [32]	Patients with intermediate or high malformations following PSARP	Mean anal resting pressure (MARP) did not differ significantly between operative groups: no significant difference in MARP between patients showing “excellent and good” outcome versus “fair and poor” outcome. Findings reported to reflect comparable long-term outcomes between traditional staged PSARP, and 1-stage neonatal PSARP.	Limited description of manometry technique/interpretation; manometry results compared with Modified Wingspread Scoring categories (e.g., “excellent”, “poor”), without definition of relationship to symptoms (e.g., constipation, FI).
Martins [50]	Patients with intermediate or high malformations following PSARP	Initial pressure, pressure after coughing, pressure after voluntary contraction, pressure after perianal stimulation, and pressure after crying were significantly higher in continent patients. Manometry findings reported to correlate with continence. Lower percentage of “normal” pressure curves in incontinent group.	Limited by technology available; study-specific continence assessment may limit comparability of findings; limited reporting of manometry findings.
Mert [55]	Children following anorectal malformation repair, able to cooperate during anorectal manometry without neurological or neurosurgical disorders	Area under curve during maximum voluntary squeeze (AUC) differed significantly between high and low malformation types. Significant relationships between resting pressure and Holschneider’s questionnaire (HQ); AUC and HQ and Rintala’s questionnaire (RQ). No correlation between function anal canal length (ACL) and Wingspread subgroups or assessed scoring systems: ACL inferred not to reflect post-operative continence status.	Small cohort in the context of the wide age range and heterogenous malformation types; limited description of cohort characteristics.
Mollard [75]	Patients with intermediate or high malformations following repair	Conscious rectal sensitivity and maximal anal resting closure pressure were normal in the majority. An RAIR was demonstrated in 50% of children with intermediate and 42.9% of children with high malformations.	Small cohort; manometry performed in sub-cohort, with group characteristics not detailed; statistical assessment of correlation between manometry findings and bowel function not reported; limited by technology available.
Nagashima [62]	Children following repair of anorectal malformations	Maximum anal pressure and anorectal pressure difference were significantly lower in high type, compared with low type, malformations. Threshold sensation pressure and maximum tolerable pressure were significantly higher in high type than low type malformations. Inadequate anal resting pressure and loss of optimal rectal sensation/reservoir function may contribute to fecal incontinence in high type malformations.	Small cohort given heterogenous nature of malformation types and patient ages; limited by technology available; study-specific scoring system lacks specificity, may limit comparability.
Niedzielski [33]	Children following PSARP	Mean values of all assessed anorectal manometry parameters were significantly lower in high versus low defect and reference groups. Mean values of all assessed parameters, except for resting anal pressure, were significantly lower in the low defect group versus the reference group.	Wide age range/time elapsed post-operatively at manometric assessment; limited description of manometric technique; statistical assessment of correlation of manometric outcomes with bowel function not reported/performed.
Ninan [71]	Children undergoing levatorplasty for management of FI following anorectal malformation repair	Best functional outcomes following levatorplasty were observed in children with good voluntary squeeze pressures on anorectal manometry prior to the operation.	Small study; limited by technology available; no description of manometry technique or equipment; limited reporting of manometry findings/interpretation.
Okada [21]	Patients following ASARP for re-operation due to FI following anorectal malformation repair	Significant increase in anal resting pressure post-operatively, with a concurrent significant increase in clinical assessment (Kelly score). Total, accident, and staining components of the Kelly score were significantly increased.	Small manometry cohort (*n =* 6); limited by available technology; assessment of correlation between manometry findings and symptoms not performed; cohort characteristics and bowel function scores provided for total cohort only.
Penninckx [16]	Infants with anorectal malformation treated at a single center	‘Normally’ functioning IAS identified after transplantation (internal sphincter-saving surgery) in supra- and translevator type defects, as well as low malformations, demonstrating the importance of IAS preservation. Anal resting tone was ‘normal’ and RAIR present in all cases.	Limited description of manometry technique and interpretation; select group underwent post-operative assessment; infants with perineal anus (surrounded by EAS) of almost normal caliber used as controls; limited by technology available.
Ray [76]	Children with intermediate or high malformation, following PSARP	Children with repaired intermediate malformations demonstrated manometric findings indicative of normal continence, with a good functional outcome. Children with repaired high malformations demonstrated grossly abnormal manometric parameters, with poor functional results.	Limited description of manometry technique, assessment, interpretation, and outcome; limited by technology available; statistical assessment of correlation between manometry findings and functional outcomes not performed/reported; limited cohort characteristics provided.
Ren [34]	Children with intermediate malformations following SILAARP or PSARP	Post-operative anal canal resting pressure was significantly higher in the SILAARP than PSARP group No significant difference between groups with respect to RAIR, high-pressure zone length, or bowel function	Manometry not performed in all patients; limited description of manometry technique and interpretation
Rintala [35]	Patients with intermediate or high malformations following repair	Mean basal anal canal pressure significantly lower in group 2 (rectal canal reconstructed from proximal rectum or sigmoid), compared with group 1 (rectal blind pouch and fistula retained during PSARP or sacroperineal pullthrough). Group 2 demonstrated significantly poorer bowel function. Group 1 demonstrated features of functional internal anal sphincter (comparable pattern of anal slow wave activity and RAIR); Group 2 demonstrated colonic type slow wave activity and no RAIR were identified.	Sedation differed between groups (Group 1 were sedated, Group 2 were not); Group 2 were older (mean 3.1 versus 8.8 years); statistical assessment of correlation between manometry findings and function not performed/reported; limited by technology available.
Rintala [36]	Patients with intermediate or high malformations	Maximum resting pressure in the anal canal was significantly higher in patients with functional internal sphincter than patients without; both were lower than the control group. The RAIR was identified in 82%, reported to suggest presence of functional internal sphincter following internal sphincter-saving PSARP.	Limited by technology available; limited description of cohort characteristics, including age at assessment and outcome with respect to function.
Rintala [58]	Patients with intermediate or high malformations	No difference in basal anal pressure or internal sphincter reflex threshold values between constipated and non-constipated patients. Anal resting pressure significantly lower in patients with soiling. Positive RAIR, considered indicative of functioning internal anal sphincter, was identified in 83%. Anal resting pressure was significantly higher in this group and a lesser proportion experienced fecal soiling (12% versus 71%).	Limited by technology available; limited description of cohort characteristics, including age at assessment; study-specific rating of bowel function may limit comparability.
Rintala [17]	Patients with intermediate or high anorectal malformations	Positive RAIR in all cases in which the distal rectal pouch was utilized in anal canal reconstruction, associated with slow pressure wave activity consistent with that of a normal anal canal. Reported to reflect functional internal sphincter in high and intermediate malformations: the distal rectal pouch with fistulous connection is an ectopic anal canal.	Small cohort; limited by technology available; limited description of cohort characteristics, including function; limited manometry data reported.
Rintala [59]	Patients following PSARP for intermediate and high malformations	Correlation between good continence outcome and the presence of a functional internal anal sphincter and a high anorectal resting pressure. Absence of functional internal sphincter, severe sacral anomalies, and constipation associated with poor functional results.	Limited by technology available; heterogenous cohort; manometry and assessment of function appear to have been performed at different time points.
Rintala [60]	Patients undergoing secondary PSARP for intractable FI following primary anorectal malformation repair	Following secondary PSARP and post-operative biofeedback training, resting and squeeze pressures significantly increased.	Small cohort; limited by technology available; limited description of manometry technique/interpretation; statistical assessment of correlation with clinical function scores not performed/reported.
Ruttenstock [18]	Patients with an externally accessible fistula	Normal presence of the RAIR identified in all pre- and post-operative rectal manometry studies. No differences between pre- and post-operative assessment with respect to RP or length of high-pressure zone.	Small cohort; limited by technology available.
Sangkhathat [63]	Infants less than three years of age, post-anoplasty for treatment of anorectal malformation	Anal resting pressure significantly higher (*p* < 0.05) and RAIR present in significantly fewer (12.5% versus 93.8%; *p* < 0.01) children with post-operative constipation than those without. No difference in RrP, ArP, RAIR, or peak squeeze pressure (PSP) between low and non-low malformation groups.	Limited characterization of bowel function; limited description of associated anomalies, including spinal anomalies.
Schuster [19]	Patients managed for perineal fistula using anal transposition technique	Maximal mean segmental pressure at rest did not differ significantly from standard results, nor did total asymmetry index; however, it was unclear whether these findings related to pre- or post-operative assessment, or a combination.	Small cohort; unclear whether post-operative manometry results were provided (assessed pre- and post-operatively).
Schuster [42]	Children with anorectal malformations following PSARP	Maximal mean segmental pressure at rest and during squeeze were reduced. Values at rest were described as pathologically low (range 6–65 mmHg), being two standard deviations below findings in healthy children (range 84–117 mmHg). No correlation between quantitative manometry outcomes and clinical score. Qualitative manometry and MRI findings were correlated; however, only a limited correlation with clinical score was identified (R = 0.425).	Small cohort; range of anorectal malformation types further restricts subgroup size.
Senel [51]	Children with anorectal malformations following repair	Significantly lower aARP in high versus intermediate anorectal malformation groups. Significantly lower aARP in good versus fair or bad continence groups.	Small, heterogenous cohort; limited description of associated anomalies (including spinal); function reported as summative scores of assessment instruments, which may limit comparability.
Sonnino [37]	Children with FI treated with gracilis muscle transposition	Continence improved post-operatively (following gracilis transposition), accompanied by an apparent trend toward great maximal pressures; however, assessment of significance not reported.	Small, heterogenous cohort; limited description of manometry outcomes; limited statistical analysis reported to aid interpretation of the significance of findings.
Tang [52]	Children with defecation disorders	Marked pressure asymmetry following PSARP. Pressure asymmetry not correlated with Rintala score.	Small anorectal malformation sub-cohort (*n =* 6); lack of normative 3D HRAM data in children.
Tong [38]	Infants with high anorectal malformation treated with LAARP vs. PSARP	Patients following LAARPT demonstrated significantly lower asymmetric index, larger vector volume, and higher anal canal pressure at rest than the PSARP group. No significant differences in length of HPZ or presence of rectoanal relaxation reflex.	Choice of intervention based on surgeon and/or parent preference (non-randomized); statistical assessment of correlation between clinical score and manometry findings not reported.
Vital Junior [53]	Children with anorectal malformation following PSARP	Overall low RP; however, RP was significantly higher in continent vs. partially continent and incontinent groups. The VCP was significantly higher in continent vs. incontinent groups. Partially continent group demonstrated VCP approaching that of the continent group, with potential positive implications for prognosis. Presence of RSR varied by group: continent: 35.5%; partially continent: 4.8%; incontinent: 6.7%.	Limited description of malformation type; idiosyncratic aspects of continence assessment may impact upon the comparability of findings.
Wang [54]	Children treated at a single center without congenital megarectum, sacral or spinal deformities	Significantly higher balloon volumes to elicit an RAIR in the anorectal malformation (vs. control) group and the low (vs. intermediate-high) defect group. An RAIR was identified in 95.7% of patients; 61.7% had “good” function. Minimum balloon volume to elicit an RAIR was negatively correlated with anal function scores.	Limited reporting of manometry findings; limited description of manometry parameters/interpretation utilized; heterogenous cohort.
Yang [39]	Children with high anorectal malformation following PSARP or LAARP	Anal canal resting pressure (ACRP) significantly higher in LAARPT group, compared with the PSARP group. No significant difference with respect to RAIR between cohorts. Absent RAIR and “lowest” ACRP reportedly observed in patients with FI.	Small sub-cohorts; association between functional outcomes and manometry findings reported without assessment of statistical correlation; relatively short follow up period and young cohort age given reported associations between anorectal manometry findings and functional outcomes.

^38^ Good prognosis: specific malformation types (rectoperineal fistula, rectovestibular fistula, imperforate anus without fistula, rectal atresia, cloaca with common channel < 3 cm); associated with a prominent midline groove, suggestive of good perineal muscle, and a normal sacrum. AA: anal atresia; aARP: average anal resting pressure; ACL: anal canal length; ACRP: anal canal resting pressure; ArP: anal resting pressure; ARPD: anorectal pressure difference; ARPG: anorectal pressure gradient; ART: anal resting tone; ASARP: anterior sagittal anorectoplasty; ASP: anal squeezing pressure; AUC: area under curve; EAS: external anal sphincter; FFI: functional fecal incontinence; FI: fecal incontinence; HD: Hirschsprung disease; HPZ: high pressure zone; HQ: Holschneider’s questionnaire; HRAM: high-resolution anorectal manometry; IAS: internal anal sphincter; LAARP: laparoscopically assisted anorectoplasty; LAARPT: laparoscopically assisted anorectal pull-through; LAR: laparoscopically assisted pull-through anorectoplasty; MARP: mean anal resting pressure; MRI: magnetic resonance imaging; MWS: modified Wingfield score; ND: neurospinal dysraphism; NI: neurological impairment; PSARP: posterior sagittal anorectoplasty; PSP: peak squeeze pressure; RAIR: rectoanal inhibitory reflex; R-APSA: Rehbein’s mucosa-stripping endorectal pull-through; RAR: rectoanal relaxation reflex; RP: resting pressure; RQ: Rintala’s questionnaire; RrP: rectal resting pressure; RSR: rectosphincteric reflex; SP: squeeze pressure; TAI: transanal irrigation; VCP: voluntary contraction pressure; -: not reported.

**Table 5 jcm-12-02543-t005:** Reported post-operative bowel function: assessment measures and outcomes.

First Author	Year	Assessment	Post-Operative Bowel Function
Arnoldi [14]	2014	Rintala score: normal (≥18), good (12–17), fair (7–11), poor (≤6). Evaluated: frequency of defecation, FI, constipation, awareness, social problems. Scores ≤ 18 classified as pathological.	Normal (≥18): 17/30 (57%). Pathological (12–17): 43%. No patient scored ≤ 11. Normal scores correlated with absence of neurospinal dysraphism and neonatal colostomy. No correlation between normal scores and malformation type or operative timing.
Banasiuk [64]	2021	Classified according to Rome III criteria into asymptomatic (A), non-retentive fecal incontinence (NRFI), constipated (C), and retentive fecal incontinence (RFI).	A: 0 (0%); NRFI: 5 (41.7%); RFI: 4 (33.3%); C: 3 (25%)
Becmeur [23]	2001	Clinical continence score. Components: ability to hold back defecation, feels the need to defecate, frequency of defecation, staining, accidents, constipation, social problems, appearance.	Group A (primary three-flap anoplasty): 16.1 (1998), 15.7 (1999) Group B (re-do three-flap anoplasty): 11.5 (1998), 15 (1999) Higher scores in children without associated anomalies: 19.6 versus 10 (*p* = 0.02). Healthy controls: 22.5
Bhat [56]	2008	Kelly score: poor (0–2); fair (3–4); good (5–6). Components: continence, staining of underwear, sphincter squeeze.	Median: 4 (range 1–6).
Burjonrappa [61]	2010	Modified Wingfield score (MWS) for fecal continence: normal function = 0, constipation/fecal incontinence = 1, intermittent fecal incontinence = 2, continuous fecal incontinence = 3.	Prior to excision of megarectum: MWS 3 Post-op: low malformations: normal (0) 6/8; (1) 1/8; 2 1/8; high malformations: normal (0) 4/6; (1) 2/6
Cahill [24]	1985	Not reported.	“Normal continence” achieved in 3/6 patients.
Caldaro [43]	2012	Modified Wingfield score: fecal continence graded as normal function (0); constipation (1); intermittent soiling >3 episodes per week (2); daily soiling (3).	Constipation: 6/17; fecal incontinence: 11/17.
Caruso [67]	2015	Abbreviated Baylor Continence Scale: score 0 (good continence)–24 (severe incontinence).	Group 1: pre-treatment 11.2 ± 0.8; post-treatment 4.7 ± 2.5 (*p* = 0.008) Group 2: pre-treatment 14.8 ± 1.1; post-treatment 7.5 ± 3.1 (*p* = 0.027) Group 3: pre-treatment 18.6 ± 1.2; post-treatment 13.3 ± 3.6 (*p* = 0.027) Group 4: pre-treatment 22.0 ± 1.8; post-treatment 17.5 ± 3.1 (*p* = 0.066)
Caruso [72]	2021	Rintala continence score: 0 (very bad) to 20 (excellent).	0 month: severe FI; 1 month: 6.14 ± 1.34; 3 months: 9.8 ± 1.57; 6 months: 16.8 ± 2.2
Chen [73]	1998	Normal: >4 years post-repair with 1–3 bowel actions daily, no FI. Mild constipation: bowel action 2–3 daily, cisapride or laxatives but without enemas, resolving with 1–2 years of conservative treatment. Severe constipation: bowel movements dependent on medication and/or enemas > 2 years post-repair. Soiling: bowel action >4 times per day, with constant or intermittent staining of underwear > 2 years post-repair.	LSARP (low anomalies). Constipation: 10/28 (35.7%); >2 years 3/25; fecal incontinence: 1/23 (4.3%); normal function: 20/23 (86.9%) PSARP (intermediate–high anomalies). Constipation: 18/25 (72.0%); >2 years 10/25; fecal incontinence: 6/22 (27.3%); normal function: 11/22 (50%) R-ASPA (intermediate—high anomalies): normal function: 5/5 (100%)
Chung [22]	2018	Bowel function score (maximum total score: 20): normal ≥18. Constipation defined according to Rome III criteria.	PSARP: median 12.5 (8–18); normal BFS 42.9%. LAARP: median 16 (10–20); normal BFS 62.5%.
Doolin [44]	1993	Clinical assessment of the following variables: (1) continence (A, incontinent; B, spotting; C, continent); (2) enema (A, none; B, occasional; C, daily); (3) rectal tone (A, none; B, fair; C, normal); and (4) sensation (A, yes; B, no).	(1) Continence: A, incontinent (7, 29%); B, spotting (9, 37%); C, continent (8, 33%) (2) Enemas: A, none (7, 29%); B, occasional (5, 20%); C, daily (12, 50%) (3) Rectal tone: A, none (1, 4%); B, fair (23, 92%); C, normal (1, 4%) (4) Sensation: A, yes (25, 100%); B, no (0, 0%)
El-Debeiky [25]	2009	Questionnaire (parent report), including Kelly Score.	Kelly score: 3–5. Fecal incontinence: 3/9.
Emblem [40]	1994	Wingspread classification: continence assessed using a rating scale (1 clean–4 constant fecal incontinence).	(1) Clean: 11; (2) staining: 3; (3) intermittent fecal soiling: 2; (4) constant soiling: 0.
Emblem [45]	1997	Continence scale (1–4): (1) continent for liquids, solids, and gas; (2) occasionally incontinent for loose stools; (3) incontinent for loose and sometimes solid stools; (4) incontinent for loose and solid stools.	Significantly better continence demonstrated following repair of low malformations (males: 1.3; 95% CI 0.7–1.8 versus females: 1.9; 95% CI 1.4–2.4) than intermediate/high malformations (3.0; 95% CI 2.5–3.5).
Fukata [41]	1997	Kelly score (max. score 6): (1) control of feces and bowel habits; (2) absence of fecal staining; (3) sling action of puborectalis. ^39^	Cohort: 4 (2–6); high malformation: 4 (3–5); intermediate malformation: 4 (2–6). Summarized as median Kelly score.
Hedlund [46]	1992	Soiling, constipation, diarrhea (grade 0–3); voluntary bowel actions.	Soiling: grade 1 (11/30); grade 1 (13/30); grade 2 (2/30); grade 3 (4/30). Constipation: grade 1 (13/30); grade 1 (8/30); grade 2 (9/30); grade 3 (0/30). Diarrhea: grade 1 (29/30); grade 1 (0/30); grade 2 (1/30); grade 3 (0/30).
Heikenen [74]	1999	-	All had FI refractory to medical therapy; however, bowel function not further defined.
Hettiarachchi [47]	2002	Modified Wingfield score	Median 3 (range 1–3).
Huang [20]	2017	Krickenbeck classification	Significantly lower incidence of FI (*p* = 0.049) and grade 2 or 3 constipation (*p* = 0.049, in the modified-PSARP group.
Husberg [26]	1992	(1) Continent; (2) minor soiling; (3) frequent soiling; (4) FI.	Positive RAIR: normal 11/15; minor soiling 3/15; frequent soiling 1/15; constipation management 10/15. Negative RAIR: minor soiling 3/6; frequent soiling 2/6; FI 1/6.
Ishihara [48]	1987	-	Translevator type: normal defecation 9/9. Intermediate type: normal defecation 2/6; constipation 2/6; incontinence 2/6. Supralevator type: normal defecation 7/23; constipation 10/23; incontinence 6/23.
Iwai [65]	1997	Kelly score	Improvement in Kelly score post biofeedback: 5/14.
Iwai [66]	2007	Japanese Study Group for Anorectal Malformation Scoring System: (1) rectal sensation; (2) constipation; (3) FI; (4) soiling. Score: good (7–8); fair (4–6); poor (0–3).	Pre-intervention: 4.5 ± 1.3 Post-intervention: 6.0 ± 1.2
Iwai [68]	1993	Kelly score	Pre-biofeedback: median 1.5 (range 0–4); post-biofeedback: median 3 (range 1–4).
Iwai [15]	1988	Kelly score	Low anomaly: median 6 (range 4–6); intermediate: 4.5 (3–6); high: 4 (1–6).
Keshtgar [69]	2008	Modified Wingfield score	Isolated anorectal anomaly: median 1 (range 1–3). Associated megarectum and/or neuropathy: median 3 (range 1–3).
Keshtgar [49]	2007	Modified Wingfield score	Pre-operative: mean 3 (range 3–3). Post-operative: mean 0 (range 0–3).
Kimura [27]	2010	Japanese Study Group for Anorectal Malformation Scoring System: (1) fecal sensation; (2) constipation; (3) FI; (4) soiling. Score: good (7–8); fair (4–6); poor (0–3).	Open anorectoplasty: poor 1/14 (7.1%); fair 10/14 (71.4%); good 3/14 (21.4%). Lap-anorectoplasty: poor 0/5 (0%); fair 4/5 (80%); good 1/5 (20%).
Kudou [28]	2005	Kelly clinical score	LAARP: 3.8 ± 1.3. PSARP: 3.4 ± 0.8.
Kumar [57]	2010	-	-
Langemeijer [29]	1991	Presence and frequency of soiling; incontinence: soiling >1/day.	Continent: 3/50; intermittent incontinence: 7/50; incontinent: 24/50; constipation: 2/50; pseudo-continence (with enemas): 20/50.
Leung [70]	2006	Soiling frequency rank; Rintala continence score.	Pre-intervention: soiling frequency rank: 1.62; mean Rintala score: 11.6 Post-intervention: soiling frequency rank: 2.59 (six month), 2.85 (twelfth month); mean Rintala score: 12.9 (six month), 13.5 (twelfth month)
Lin [30]	1996	(1) Continent with constipation; (2) continent without constipation; (3) occasional soiling (<3/week); (4) incontinent (>4/week).	
Lin [31]	2003	Frequency of bowel actions: <1/day; 1–4/day; >5/day.	PSARP: <1/day 3/13 (23%); 1–4/day 7/13 (54%); >5/day 3/13 (23%). LAR: <1/day: 1/9 (11%); 1–4/day 7/9 (78%); >5/day 1/9 (11%).
Liu [32]	2004	Modified Wingspread score: excellent–poor outcome. Constipation: <3 bowel actions/week, rectal impaction, or fecal mass. Continence: ability to stay clean without staining or soiling. FI: regular loss of solid feces. Soiling: loss of small amounts of feces.	Group 1 (PSARP): constipation 23/48; soiling/FI: 23/48. Good–excellent score: 28/48 (58.3%). Group 2 (1-stage PSARP): constipation 29/65; soiling/FI: 33/65. Good–excellent score: 35/65 (53.8%)
Martins [50]	1996	Clinical + manometric assessment: (1) continent: 1–2 bowel actions/day without FI, good upper and lower rectal contractions; (2) partially continent: 3–5 bowel actions/day with frequent soiling or rectal prolapse, with moderate upper or lower rectal contractions; (3) incontinent: >5 bowel actions/day, constant FI, large anal opening, and light or no upper or lower rectal contractions.	Continent: 13/27 (1 with sacral malformation, 12 without sacral malformations). Partially continent: 7/27 (3 with sacral malformations, 4 without sacral malformations). Incontinent: 7/27 (6 with sacral malformations, 1 without sacral malformation).
Mert [55]		Holschneider’s questionnaire (HQ); Rintala’s questionnaire (RQ); Krickenbeck’s questionnaire (KQ), and Peña’s questionnaire (PQ).	Summary statistics not provided (see paper, Table 2).
Mollard [75]		Clinical assessment: subjective parent report.	Fecal incontinence: 1/13 (cloacal malformation); constipation resolving within 1 post-operative year: 12/13.
Nagashima [62]		Clinical classification: (1) normal bowel function; (2) FI: incontinence >2/week; (3) constipation: enema required daily to achieve bowel action.	Normal: 18/32 (5 high malformation, 4 intermediate malformation, 9 low malformation) FI: 12/32 (10 high malformation, 2 intermediate malformation) Constipation: 2/32 (high malformation).
Niedzielski [33]		Modified Kelly score. Assessed: frequency of stools (0–2 pts); stool consistency (0–2 pts); staining (0–2 pts); need for treatment (drugs, enemas, diapers) (0–2 pts); rectoanal inhibitory reflex on manometry (0 or 2 pts). Score: good (10–8); satisfactory (7–4); poor (0–3).	Good: 83/91 (91.2%); fair 5/91 (5.5%); unsatisfactory 3/91 (3.3%).
Ninan [71]	1994	Clinical assessment (excellent, improved, poor). Criteria not reported.	Pre-levatorplasty: persistent FI in all patients. Post-levatorplasty: excellent 6/13 (46%); improved 5/13 (39%); poor 2/13 (15%).
Okada [21]	1993	Kelly score: poor (0–2); fair (3–4); good (5–6).	Poor: 7/10; fair: 3/10; good: 0/10.
Penninckx [16]	1986	-	-
Ray [76]	2004	Kiesewetter criteria	Intermediate anomalies: continent. High anomalies: poor continence in 50%.
Ren [34]	2019	Krickenbeck classification	Comparable rates of voluntary bowel movements, constipation, and FI between SILAARP and PSARP groups.
Rintala [35]	1990	Clinical assessment: degree of soiling; medication requirement. (1) Good: voluntary bowel movements without medication or FI; (2) fair: minor FI or clean with oral medication; (3) poor: daily FI or enema dependent.	Group 1 (PSARP, Stephens): good (6/14); fair (4/14); poor (4/14). Group 2 (sacroabdominoperineal ± PSARP redo): good (0/16); fair (10/16); poor (6/16).
Rintala [36]	1995	-	-
Rintala [58]	1993	Clinical assessment of constipation, soiling, voluntary bowel movements. Constipation graded 0 (no constipation)–3 (severe, unmanageable).	Constipation requiring management: 26/40 (65%). FI: 4/33 (12%) with functioning IAS (determined by RAIR); 5/7 (71%) without.
Rintala [17]	1990	-	-
Rintala [59]	1995	Clinical scoring of anorectal function, determined from hospital records. Maximum score = 20.	Excellent: 16/46 (35%; mean 18.4 ± 0.6). Good: 16/46 (35%; mean 13.2 ± 2.0). Fair: 10/46 (22%; mean 8.8 ± 1.4). Poor: 4/46 (9%; mean 7.5 ± 1.7). Intermediate vs. high malformation: mean 16.1 ± 4.2 vs. mean 12.9 ± 3.9.
Rintala [60]	1995	Modified Holschneider score. Components: frequency of defecation, stool consistency, soiling, sensation, urgency, discrimination of bowel contents, need for care. Poor (0–4); fair (5–9); good (10–14).	Pre-operative: 7.8 ± 1.3. Post-operative: 10.1 ± 1.6.
Ruttenstock [18]	2013	Krickenbeck outcome assessment.	All patients demonstrated voluntary bowel movements with no FI. Constipation (Krickenbeck grade 2): 3/12.
Sangkhathat [63]	2004	Patients classified as constipated (<1 bowel action/day, laxative/enema requirement and/or grey discoloration or hard feces) or not constipated.	Constipation: 8/24 (33%).
Schuster [19]	2000	-	-
Schuster [42]	2001	Modified Kelly score (maximum total score 14): frequency of bowel movements, stool consistency, soiling, sensation, feeling of fullness, warning period, need for bowel care	Good (10–14): 4/17; fair (5–9): 6/17; poor (0–5): 7/17.
Senel [51]	2007	Kelly clinical score and Kiesewetter-Chang score (good: continent; fair: socially continent; poor: incontinent).	Kelly score: median 5 (range 0–6). Kiesewetter-Chang: good 14/18 (78%); fair 3/18 (17%); poor 1/18 (5%).
Sonnino [37]	1991	Clinical evaluation: (1) soiling, (2) necessity for diapers, (3) dietary requirements.	Continent: 4/7; occasional soiling (<2 episodes/week): 3/7.
Tang [52]	2017	Rintala score	Mean 18.3 (range 17–20).
Tong [38]	2011	Kelly score. Components: FI, fecal staining, sphincter squeeze.	LAARPT: 3.52 ± 1.42. PSARP: 3.49 ± 0.82
Vital Junior [53]	2007	Continent (1–2 bowel actions/day, no FI, good sphincter contractility); partially continent (3–5 bowel actions/day, frequent FI, upper and lower sphincter contractility only); or fecally incontinent (>5 bowel actions/day, constant FI, abnormal anal appearance absent or weak contractions of the sphincter complex).	Continent: 31 (37.8%); partially continent: 21 (25.6%); incontinent: 30 (36.6%).
Wang [54]	2016	Scoring system, including (1) desire to defecate, (2) constipation, and (3) FI. Anal function rated as: good (5–6 points), intermediate (3–4 points), or poor (0–2 points).	Good (5–6 points): 29 children (61.7%); intermediate (3–4 points): 13 children (27.7%); and poor (1–2 points): 5 patients (10.6%).
Yang [39]	2009	Kelly score. Components: continence, staining, squeezing force.	LAARPT: 3.91 ± 1.14 (good: 4; fair: 5; poor: 2). PSARP: 3.83 ± 1.40 (good: 4; fair: 6; poor: 2).

^39^ Method of assessment of puborectalis muscle sling action not defined. C: constipated; FI: fecal incontinence; HQ: Holschneider’s questionnaire; IAS: internal anal sphincter; KCS: Kelly’s clinical score; KQ: Krickenbeck questionnaire; KS: Krickenbeck score; LAARP: laparoscopically assisted anorectoplasty; LAARPT: laparoscopically assisted anorectal pull-through; LAR: laparoscopically assisted pull-through anorectoplasty; LSARP: limited sagittal anorectoplasty; MWS: modified Wingfield score; NRFI: non-retentive fecal incontinence; PQ: Peña’s questionnaire; R-ASPA: Rehbein’s mucosa-stripping endorectal pull-through in combination with anterior sagittal perineal anorectoplasty; RFI: retentive fecal incontinence; RQ: Rintala’s questionnaire; (-): not reported.

## Data Availability

No new data were created or analyzed in this study. Data sharing is not applicable to this article.

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
