# Peer review of "Post-Operative Anorectal Manometry in Children following Anorectal Malformation Repair: A Systematic Review"

_jcm, 2023, doi:10.3390/jcm12072543_

Round 1

Reviewer 1 Report

Evans-Barnes et al. present (1) a systematic review of existing data regarding anorectal manometry results in children following anorectal malformation repair, and (2) evaluate the manometry protocols utilized, including equipment, assessment approach, and interpretation.

The amount of cases and data collected is impressive (n>2000). The methods are well described and the results are clearly presented. However, the main limitation of the study is the heterogeneity of the data included. This is true on multple levels. The age range of patients and time of follow up are both very variable. A wide range of anorectal interventions were included, and the surgical technique (introduced in 1982) has changed over time with significant progress made in the last 20 years. Moreover, the indication for anorectal manometry was not consistent between studies.

In conclusion, this paper may be an important collection of clinical data; however, as the authors recognize, it is difficult to make any conclusions not only due to lack of standardization in the anorectal manometry but also the heterogeneity of the patients included in the analysis. The appeal to standardization, as has recently been established in the performance of anorectal manometry in adults (London Classification) is very appropriate. 

Author Response

Thank you for your comments. In this review we aimed to synthesize the methodology (equipment, protocols, interpretation) and outcomes of anorectal manometry performed in children following anorectal malformation repair. In doing so, we sought to evaluate our current understanding of anorectal function in this cohort to guide future work. We agree that heterogeneity is the prevailing limitation of studies performed in the assessed cohort.

With respect to the time frame selected, the posterior sagittal anorectoplasty (PSARP), described in 1982, revolutionized both the operative approach to anorectal malformations and understanding of the continence mechanism. Improved outcomes with respect to bowel function have been demonstrated in the PSARP era. Although this systematic review was not designed to assess only outcomes following PSARP, we elected to assess anorectal manometry findings in the period following this seminal conceptual shift, allowing a three-year period for its wider adoption. More targeted systematic reviews may facilitate a more homogenous patient population, with respect to age, type of surgical repair, or indication for manometry. However, as this systematic review has demonstrated, inconsistent inter-center manometry practice, interpretation, and reporting will continue to impede meaningful comparison of findings.

As outlined in the Discussion and Conclusion, there is an urgent need for standardization across multiple aspects of anorectal manometry performance and reporting. Consistent application of manometry protocols, such as the London Classification as developed for adults, is essential to improving study consistency between centers. However, while a standardized protocol is essential, so too is the standardized reporting of cohort characteristics and clinical outcomes; without this we cannot compare findings between studies. We hope that highlighting all of the existing limitations, through reviews such as this, will help to improve the quality and consistency of work performed in this population more broadly.

Reviewer 2 Report

The authors present a well performed review on post-operative anorectal manometry in patients with anorectal malformations.

The inclusion and exclusion criteria a well described. The selection of included articles is correct. 

Overall, there is still a large heterogenity of the reported results. The manin concern is, that some studies included all patients postoperatively, some only those who had ongoing continence problems. 

Further, the authors did not reflect the influence of the specific form of ARM (i.e. high) on the manometry findings, which would be of interest to the readers. Also, a clear differentiation between patients with and without neurogenic bowel is missing.

The form of ARM usually corresponds with the method of surgical repair. The authors should comment on that.

Author Response

  1. The authors present a well performed review on post-operative anorectal manometry in patients with anorectal malformations. The inclusion and exclusion criteria a well described. The selection of included articles is correct. 

Overall, there is still a large heterogeneity of the reported results. The main concern is, that some studies included all patients postoperatively, some only those who had ongoing continence problems.

Thank you for your comments. We agree that heterogeneity – affecting multiple aspects of the included studies – is the prevailing limitation of work performed in the assessed cohort. As outlined in the Discussion and Conclusion, there is an urgent need for standardization across multiple aspects of anorectal manometry performance and reporting.

We agree there is significant heterogeneity in the reported symptoms experienced by the included cohorts. As you have highlighted, some studies included all patients post-operatively, whilst others included only those with post-operative bowel dysfunction. Further to this, other studies did not clarify the bowel function of the target and/or included population. Adding further complexity, few studies provided symptom definitions or diagnostic criteria; of those that did, the approach to bowel function assessment and diagnosis of dysfunction varied markedly. For example, constipation was defined as “less than three bowel actions per week”, “less than one bowel action per day”, and “enema required daily to achieve bowel action”[1-3].Consequently, comparing within reported diagnostic groups (i.e. “constipation”) is likely to be inaccurate.

Consistent application of manometry protocols and study interpretation (including definitions of “normal”) are essential to improving study consistency between centers. Furthermore, we believe standardized reporting of cohort characteristics and clinical outcomes is crucial. Standardized clinical outcome reporting would facilitate the comparison of findings between centers and the identification of manometry-symptom correlates. Until this is achieved, furthering our understanding of anorectal function in this group, both with and without post-operative bowel dysfunction, remains challenging. We hope that highlighting the existing limitations, through reviews such as this, will help to improve the quality and consistency of work performed in this population more broadly.

  1. Further, the authors did not reflect the influence of the specific form of ARM (i.e. high) on the manometry findings, which would be of interest to the readers.

Thank you for this recommendation. Given the variability across nearly all aspects of study conduct (equipment, protocol, parameter definitions, study interpretation), comparison of findings between studies by anorectal malformation type was not performed. Where possible throughout, reference has been made to within-study findings related to specific manometry parameters, reported by anorectal malformation type. For example, “More complex malformations were typically associated with lower resting pressures than less complex malformations [2,4-11], although this was not uniform [3].”

A corresponding sentence has been added to the Results section “Squeeze pressure”: “Of those that reported comparison of squeeze pressure by complexity of malformation type, there was some evidence that more complex malformation types had reduced squeeze pressures relative to less complex types [2,12]; however, this was not uniform [13].”

  1. Also, a clear differentiation between patients with and without neurogenic bowel is missing.

Thank you. We agree this a fundamental limitation of the current literature. Anorectal malformations frequently occur as part of the VACTERL association, which may include comorbid sacrospinal anomalies. Details regarding associated malformations were provided by fewer than half of the included studies (44.4%). Of those that did, not all reported the proportion with associated sacrospinal anomalies, whilst even fewer reported their form or severity. As our systematic review concludes, the regulation of defecation is multifactorial; the prevailing limitation of our current understanding of bowel dysfunction following anorectal malformation repair is the failure to place manometry findings into the broader context of the continence mechanism. As highlighted on page 45, standardized reporting of clinical characteristics should include documentation of associated anomalies, particularly those of the sacrum and spine.

  1. The form of ARM usually corresponds with the method of surgical repair. The authors should comment on that.

Thank you for highlighting this important point. Type of anorectal malformation is an important consideration when determining the most appropriate approach to operative repair. Whilst this review was not designed to compare post-operative anorectal manometry findings between surgical repair types, this would be an important factor to consider in future repair-focused reviews. As with other factors highlighted in this review, meaningful comparison would be dependent upon adequate reporting of both of these characteristics (anorectal malformation type and approach to repair), along with other determinants of continence relevant to this population (such as sacrospinal integrity), and placed within the context of the cohort’s post-operative bowel function. Figure 2 has been added to highlight these key cohort characteristics that should be reported.

A systematic review focused upon comparison of operative approaches may facilitate a greater understanding of manometry findings between repair types; however, as this systematic review has demonstrated, it is likely that the current inconsistent manometry practice, interpretation, and reporting would limit meaningful comparison of findings. Minimum reporting standards should be developed to support this process.